©c Author(s) 2023. CC BY 4.0 License.



# Extension, Development and Evaluation of the representation of the OH-initiated DMS oxidation mechanism in the MCM v3.3.1 framework

Lorrie S.D. Jacob[1], Chiara Giorio[1], and Alexander T. Archibald[1,2]

[1]Yusuf Hamied Department of Chemistry, University of Cambridge, Cambridge, CB2 1EW, UK
[2]National Centre for Atmospheric Science, Cambridge, CB2 1EW, UK

**Correspondence:** Lorrie S.D. Jacob (lj384@cam.ac.uk), Alexander T. Archibald (ata27@cam.ac.uk)

**Abstract.** Understanding dimethyl sulfide (DMS) oxidation can help us constrain its contribution to Earth's radiative balance. Following the discovery of hydroperoxymethyl thioformate (HPMTF) as a DMS oxidation product, a range of new experimental chamber studies have since improved our knowledge of the oxidation mechanism of DMS and delivered detailed chemical mechanisms. However, these mechanisms have not undergone formal intercomparisons to evaluate their performance.

This study aimed to synthesise the recent experimental studies and develop a new, near-explicit, DMS mechanism, through a thorough literature review, as a reference mechanism for future work to build on. A simple box model was then used with the mechanism to simulate a series of chamber experiments, and evaluated through comparison with four published mechanisms. Our modelling shows that the mechanism developed in this work outperformed the other mechanisms on average when compared to the experimental chamber data, having the lowest fractional gross error for 8 out of the 14 DMS oxidation products

studied. A box model of a marine boundary layer was also run, demonstrating that the deviations in the mechanisms seen when comparing them against chamber data are also prominent under more atmospherically relevant conditions.

Although this work demonstrates the need for further experimental work, the mechanism developed in this work, having been evaluated against a range of experimental conditions, provides a good basis for a near-explicit DMS oxidation mechanism that would include other initiation reactions (e.g., halogens), and can be used to compare the performance of reduced mechanisms

used in global models.

## 1   Introduction

Dimethyl sulfide ($CH_3SCH_3$, DMS) is the largest natural source of sulfur in the atmosphere (Bates et al., 1992). It is formed from phytoplankton when they undergo physiological stress (Hopkins et al., 2023) and to a smaller extent from terrestrial vegetation (Vettikkat et al., 2020). The presence of a sulfur atom in the molecule leads to rather complex oxidation mechanisms,

much more complex than similar sized hydrocarbons (e.g., Calvert (2008)). The earliest study on DMS OH-initiated oxidation dates back to the 1970s (Cox and Sandalls, 1974), the decade before DMS was postulated as being involved in a global homeostatic cycle later termed the CLAW hypothesis (Charlson et al., 1987). This (commonly refuted) hypothesis that warmer temperatures would cause phytoplankton to emit more DMS, resulting in higher concentrations of cloud condensation nuclei



(CCN) and an increase in cloud formation, effectively counteracting global warming, has since led to an abundance of research

(Ayers and Cainey, 2007). Barnes et al. (2006) performed a comprehensive overview of the oxidation of DMS, capturing the major literature up-to ca. 2006. The review from Barnes et al. (2006) highlights the major features of DMS oxidation 1) OH is the principle oxidant of DMS and can initiate oxidation via H-atom abstraction at a methyl group or OH-addition to the sulfur atom 2) the nitrate radical ($NO_3$) primarily reacts via H-atom abstraction 3) halogen atoms (Cl, Br, I) and halogen oxides (ClO, BrO, IO) can undergo H-atom abstraction, halogen-atom addition and O-atom addition to the sulfur atom to

form dimethyl sulfoxide (DMSO). Based on these initial oxidation mechanisms a range of oxidation products are possible.

In spite of the extensive previous work on DMS, several aspects of its oxidation have remained highly uncertain and new surprises have emerged in the last few years. One of the major oxidation products of DMS is the methylthiomethyl peroxy radical ($CH_3SCH_3OO$, MTMP). Based on *ab initio* calculations, Wu et al. (2015) suggested that this could undergo atmospheric autoxidation and generate hydroperoxymethyl thioformate (HPMTF); a mechanistic pathway that was not considered

previously in the Barnes et al. (2006) review. The existence of HPMTF in the atmosphere was conclusively established thanks to global aircraft observations (Veres et al., 2020). Additionally, recent work suggests that the formation and deposition of a major product of DMS oxidation, methanesulfonic acid ($CH_3SO_3H$, MSA), is not accurately modelled (Chen et al., 2018; Hoffmann et al., 2016, 2021). MSA has been known to contribute to cloud condensation nuclei (CCN) due to its low volatility and hygroscopicity (Charlson et al., 1987; Ayers et al., 1996; Curry and Webster, 1999), making the accurate modelling of the

compound important to understand the role of DMS oxidation in cloud formation.

New updates to our understanding of DMS oxidation could have important impacts on our understanding of the role of DMS in the climate system. DMS sets the natural sulfur background in most parts of the world and, as has been described previously, could play an important role in CCN. Fung et al. (2022) looked at the effects of updates to the gas and aqueous phase DMS oxidation on radiative forcing using the CAM6-Chem/CESM2 model. They found that updates to gas-phase chemistry,

including HPMTF, led to significant changes in the pre-industrial aerosol burden and so the change in radiative forcing from the pre-industrial to the present day. However, these changes were counteracted when they accounted for updates to the aqueous phase chemistry of DMS.

The mechanism of Fung et al. (2022) was a significant improvement from the standard chemistry used in CAM6-Chem/CESM2 for DMS (similar for other CMIP6 era Earth System Models e.g., Cala et al. (2023)). However, the DMS mechanism itself was

never evaluated against experimental chamber data. Instead, the authors evaluated the performance of the scheme through comparison of measured and simulated species related to DMS, an approach that can be affected by emission bias. Fung et al. (2022) used as their emissions of DMS the Lana et al. (2011) sea surface DMS climatology. Whilst this is widely regarded as a reference DMS climatology, the simulation of atmospheric DMS by Fung et al. (2022) was significantly high biased compared to the median of aircraft and ground based observations.

The best way to improve our understanding of DMS oxidation, and by proxy its impacts on the climate system, is through the development of comprehensive mechanisms evaluated against experimental data and the incorporation of these mechanisms into complex models. There have been a number of experiments recently and a number of detailed mechanisms have been developed. Ye et al. (2022), Jernigan et al. (2022) and Shen et al. (2022) performed simulation chamber experiments under





a wide range of conditions (mixing ratios of reagents, photolysis environments, temperature, humidity) and modelled their

experiments using near-explicit mechanisms that included the HPMTF pathway, along with adjustments to the DMS subset of the Master Chemical Mechanism (MCM v3.3.1). In the case of the Shen et al. (2022) study, their mechanism was tied to the Hoffmann et al. (2016) mechanism, which led to more deviations from the MCM. The MCM is often used as a starting point for mechanism development, however, the MCM DMS scheme suffers from a number of problems. Firstly, unlike the other VOCs simulated by the MCM (alkanes, alkenes, aromatics and oxygenates), the DMS scheme never underwent evaluation

against chamber experiments. Secondly, the MCM DMS scheme is rather outdated. It fails to account for the autoxidation of MTMP and the sulfur chemistry updates from the literature captured in the most recent NASA panel review (Burkholder et al., 2019).

Moreover, when mechanisms have been developed they have generally been developed based on one or a small subset of experimental chamber studies. This is especially true for the recent studies focused on DMS that have solely developed

mechanisms based on individual chamber studies. Given that the settings of a chamber (its volume, material, and its inputs of gases) vary significantly, it's unlikely that each independent chamber study has developed a DMS mechanism under the same set of conditions. There have also been no studies that synthesise and intercompare these detailed mechanisms.

In this study we use the MCM DMS mechanism as a template to develop a near-explicit gas phase DMS mechanism, focusing on the OH-initiated chemistry. The newly developed mechanism was compared against recently developed mechanisms that

have also, largely, been based on the MCM. We intercompared the results of mechanisms reported in the literature against each other and against numerical simulations of the corresponding array of chamber experiments they were derived from. In this paper, we outline the mechanism generated in this study and the other mechanisms used. We outline the chamber experiments that all mechanisms were evaluated under, along with our evaluation metrics, and the performance of the mechanisms. We detail the results for some key DMS oxidation products, and discuss the key atmospheric implications and conclusions of the

study.

## 2 Evaluated mechanisms

In this study, four published DMS mechanisms were compared to each other, and to results from experimental chamber studies. The mechanisms were extracted from the primary studies as: Ye et al. (2022), Jernigan et al. (2022), Shen et al. (2022) and the MCM v3.3.1 (Saunders et al. (2003), hereafter Ye, Jernigan, Shen and MCM). In addition to the four mechanisms from the

literature that were evaluated, we also developed a new DMS mechanism for this study. A comparison of the major reactions included in the five mechanisms evaluated in this paper is given in Figure 1.

Only the sulfur reactions from the mechanisms were compared in this study. In addition to the MCM, the Ye mechanism included 12 reactions of the HPMTF pathway. The Jernigan mechanism added only five new reactions, and adjusted four existing reactions. Most of the Jernigan et al. (2022) changes involved adding a simplified HPMTF pathway, along with

changes to the methylthiomethyl peroxy ($CH_3SCH_2O_2$, MTMP) radical reaction with $RO_2$, and the DMSO reaction with an OH radical. These changes to the MCM were based on the mechanism file included in the supplementary data of the Jernigan





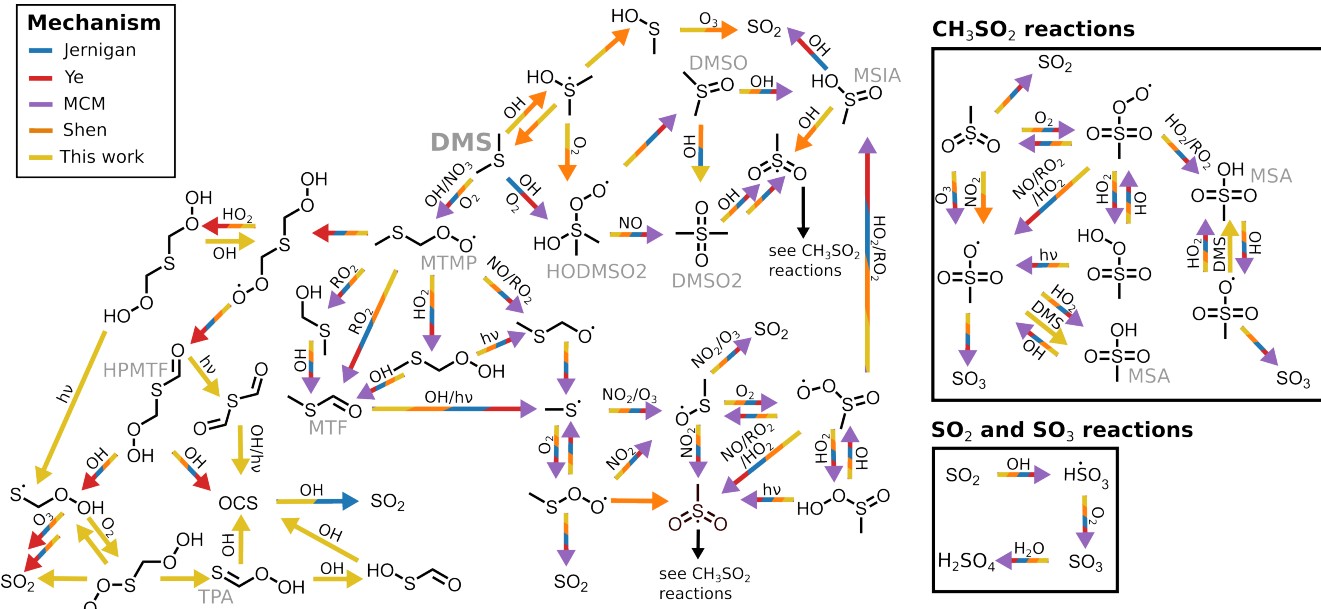

**Figure 1.** The major reactions of the DMS oxidation pathway for the five mechanisms discussed in this work, with the colours of the arrows representing the mechanisms that included the respective reactions; Jernigan (blue), Ye (red), MCM (purple), Shen (orange), and the mechanism developed in this work (yellow). Only the sulfur products are included. Note that not every reaction that was included in the mechanisms is shown (see Appendix Table 1 and supplementary Section S1 for full details).

et al. (2022) paper, which differed slightly from the table of reactions given in their supplementary information. The Shen mechanism was based on the mechanism developed in Hoffmann et al. (2016), and as such has the largest deviation from MCM; 25 sulfur reactions were added, and 4 adjusted (compared to the MCM). In addition, there were 19 non-sulfur reactions

added, adjusted or removed, but these reactions were not included as part of the Shen mechanism used in this study, only in the auxiliary file for the Shen et al. (2022) experiment (this is explained in further detail in Section 3.1). The reactions adjusted or added to the MCM by the different mechanisms are given in the supplementary information (Section S1).

The mechanism developed in this study was based on a thorough literature review, to update and improve the DMS mechanism in the MCM. To determine which rate constants should be used in the model, the same construction methodology as

the MCM (Saunders et al., 2003) was used. This methodology prioritises evaluated data (such as the NASA panel report (Burkholder et al., 2019) and IUPAC (Atkinson et al., 2004)), followed by published experimental data, structure-activity relationships, and theoretical calculations, respectively. In some cases where rate constants had not been experimentally determined, we manually adjusted them to improve the performance of the mechanism in the chamber studies. One exception was the decomposition of $CH_3SO_2$, for which the rate constant for the reaction had been experimentally determined, but there

was no consensus in the literature (described in more detail in Section 5.4), so it was also adjusted for this study. Overall,



62 reactions were added and 21 reactions were adjusted from the base MCM mechanism. A more detailed explanation of the construction of this mechanism, along with tables of the reactions added and adjusted, are included in the Appendix.

# 3 Experiments studied

The experiments studied in this work were conducted under a range of different experimental conditions, and by extension model input parameters. These experimental conditions are summarised in Table 1. To evaluate the mechanisms against the different experiments, a consistent approach was taken to deal with the different methods of modelling the experiments taken by the authors of the papers.

**Table 1.** A summary of the experimental conditions of the experiments studied in this paper (Albu et al. (2008); Ye et al. (2022); Jernigan et al. (2022); Shen et al. (2022)). The $RO_2$ and OH radical concentrations were found through the box modelling of the experiments (using the mechanism developed in this work).

| Experiment | Albu et al. (2008) | Ye et al. (2022) Exp. 1 | Ye et al. (2022) Exp. 2a | Jernigan et al. (2022) | Shen et al. (2022) |
|---|---|---|---|---|---|
| Temp (K) | 290 | 295 | 295 | 298 | 263 |
| Chamber ($m^3$) | 0.34 | 7.5 | 7.5 | 0.6 | 26.1 |
| OH Source[a] | $H_2O_2$ | HONO | $H_2O_2$ | TME + $O_3$ | $O_3$ (+ $H_2O$) |
| Avg. OH ($cm^{-3}$) | $2.9 \times 10^7$ | $5.9 \times 10^6$ | $1.5 \times 10^6$ | $1.4 \times 10^6$ | $5.7 \times 10^6$ |
| Avg. $RO_2$ ($cm^{-3}$) | $2.0 \times 10^{11}$ | $1.6 \times 10^7$ | $4.7 \times 10^8$ | $1.8 \times 10^9$ | $3.2 \times 10^8$ |
| RH (%) | - | 1 | 1 | <0.5 | 70 |
| DMS (ppb) | 15000 | 72.8 | 82 | 10 | 0.6 |
| NO (ppb) | - | 50 | - | - | - |
| $NO_2$ (ppb) | - | 90 | - | - | - |
| $H_2O_2$ (ppb) | 25000 | - | 1500 | - | - |
| HONO (ppb) | - | 90 | - | - | - |
| CO (ppb) | - | - | - | - | 120 |
| $O_3$ (ppb) | - | - | - | 23 | 125 |
| VOC photolysis | No | Yes (300-400 nm) | Yes (300-400 nm) | No | No |
| Duration (h) | 0.5 | 2 | 5 | 20 | 6 |

[a] apart from the Jernigan et al. (2022) experiment, OH was formed through the photolysis of the given precursor

## 3.1 Modelling of the experiments

The zero-dimensional box model BOXMOX was used in this work (Knote et al., 2015), an open-source wrapper for the Kinetic
PreProcessor (KPP) (Sandu and Sander, 2006) software.



When first modelling an experiment that had been modelled by the respective authors, the input parameters and mechanism from that paper were used. This allowed us to directly compare our modelled output to their modelled outputs, and ensure we were able to correctly replicate the chemical system. Figure 2 shows our replication of the output from these papers was in generally excellent agreement (compare the black solid and coloured dashed lines). In replicating the modelled outputs

from the papers, some of our model outputs deviated by up to 16% at the end of the experiment, however, the larger deviations tended to be for the minor species, and in all cases were well within the deviations between the different mechanisms evaluated, providing confidence that we were able to faithfully simulate using a unified framework, the different experimental chamber studies from the Ye et al. (2022), Jernigan et al. (2022) and Shen et al. (2022) papers.

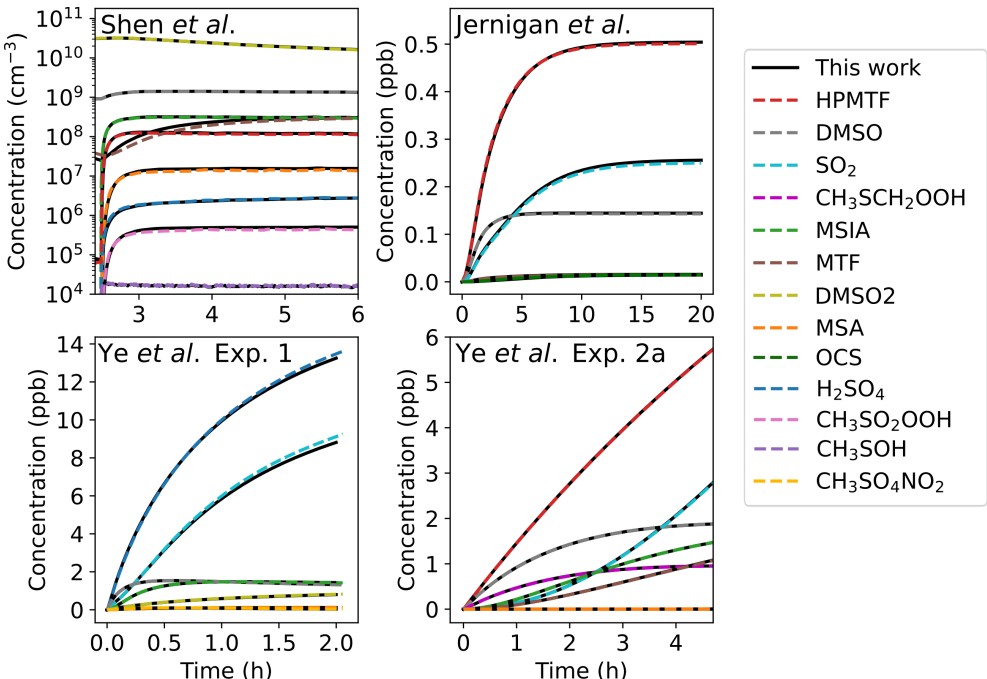

**Figure 2.** Reproduction of the modelling results from the chamber experiments simulated in this study. The modelling of the experiments from the authors of Shen et al. (2022), Jernigan et al. (2022) and Ye et al. (2022) are shown in coloured dashed lines (obtained via request or through supplementary data attached to the respective paper), our modelling using the same mechanism and input files as given are shown in black solid lines.

It is worth noting that special treatment was needed in modelling the Ye et al. (2022) experiment 1 and Albu et al. (2008)

experiment. In the case of the Ye et al. (2022) experiment 1, the input concentration of $NO_2$ was increased to reflect the measured concentration, which resulted in less DMS being consumed. See the supplementary (Sections S2 and S3) for details.

When comparing the mechanisms to the experiments, we only wanted to compare the effects of changing the gas-phase sulfur reactions. To do this, we kept the model inputs and mechanism consistent between the simulations of an experiment, with the exception of the sulfur reactions adjusted between the mechanisms. Jernigan et al. (2022) and Shen et al. (2022) also adjusted





non-sulfur reactions, such as the reaction between methyl peroxy ($CH_3O_2$) and OH radicals. These non-sulfur reactions were kept in the auxiliary mechanism of an experiment to keep our modelling of an experiment from a paper consistent with the modelling in that paper by the authors. In addition, reactions for the loss from dilution, wall loss, and in the case of Shen et al. (2022), heterogeneous wall reactions (forming dimethylsulfoxide, DMSO and dimethyl sulfone, DMSO2), were also included by us in the auxiliary mechanisms for each experiment. In the auxiliary mechanism for the Jernigan et al. (2022) experiment, the tetramethylethylene (TME) subset of the MCM was included, with the adjustments from the authors. Links to these auxiliary mechanisms can be found in the data availability section.

## 3.2 Comparison of the mechanisms

After the input parameters and auxiliary mechanisms for the experiments had been determined, the different mechanisms were compared to the chamber studies. This was done for all experiments and is included in the supplementary information (Section
S4), with the Jernigan et al. (2022) experiment results shown here as an example.

The Jernigan et al. (2022) experiment was performed in a low $NO_x$ environment, with OH produced from the ozonolysis of tetramethylethylene (TME). The seven products measured in the experiment are shown in Figure 3, with the modelling outputs of the Jernigan, Shen, Ye and MCM mechanisms, along with the mechanism developed in this study.

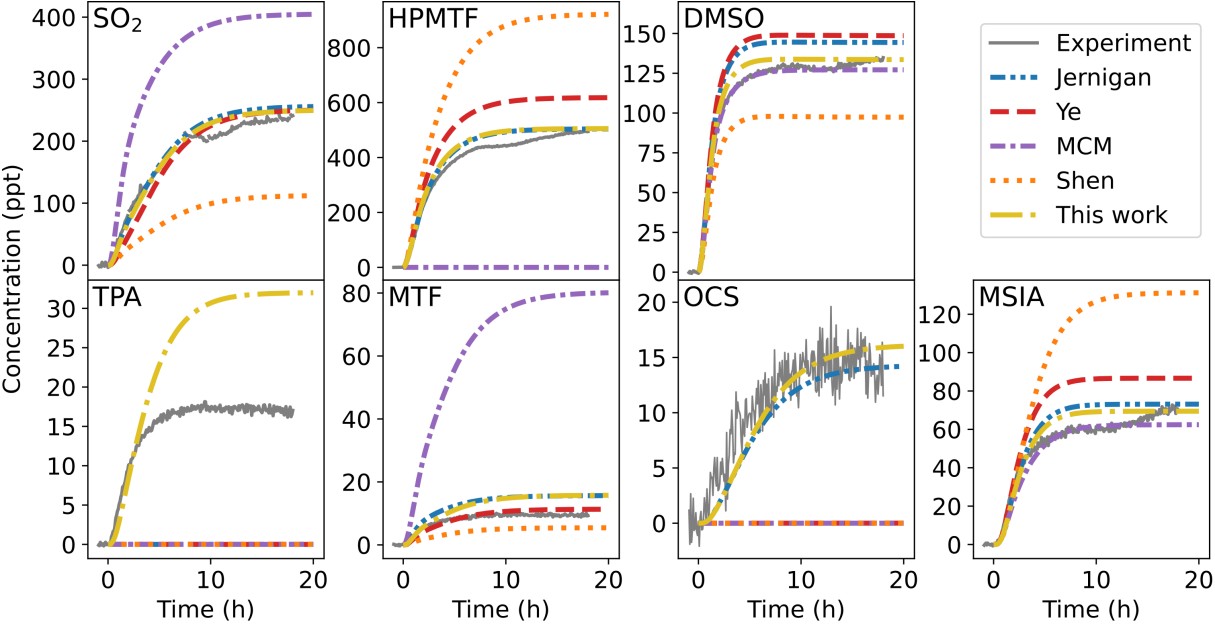

**Figure 3.** Comparison of the products measured in the Jernigan et al. (2022) experiment (grey solid lines), to our modelling results using the Jernigan (blue dot dot-dashed lines), Ye (red dashed lines), MCM (purple dot-dashed lines) and Shen mechanisms (orange dotted lines), along with the mechanism developed in this work (yellow dot-dashed lines).



As shown in Figure 3 (and Figures S3-S11 in the SI), the performance of the DMS mechanisms deviated greatly from each
other, and the experiment(s), especially in the low $NO_x$ conditions of the Jernigan et al. (2022) experiment. Some products,
such as thioperformic acid (TPA), were not included by most mechanisms, while others, such as carbonyl sulfide (OCS), were
produced in much lower concentrations by some of the mechanisms than was observed experimentally.

### 3.3    Evaluation metrics used

The mechanisms in this paper are evaluated on the basis of three metrics: fractional gross error (FGE), modified mean bias
(MMB) and correlation ($\rho$). These metrics are normalised and thus independent of the units used, or the relative intensities
of the species. This makes it easier to compare across different experiments, which use different units, and a large range of
precursor concentrations (the initial DMS concentration used in the experiments ranged from 0.6-15000 ppb). The measured
and modelled concentration over the time steps of the experiment were utilised for these calculations, and the modelled outputs
were interpolated to the observation time steps, with any gaps in the observations not included in the analysis.

The MMB (given in the equation below), is a normalised version of mean bias. This metric provides the bias between the
model and the observations within a range of -2 (negative bias) to 2 (positive bias), with 0 being ideal, where the model matches
the observation.

$$MMB = \frac{2}{n}\sum_i \frac{M_i - O_i}{M_i + O_i} \tag{1}$$

where $M = (M_i)_{0 \leq i \leq n}$ is a vector of modelled values and $O = (O_i)_{0 \leq i \leq n}$ is a vector of the observed (experimental) values.

The FGE is the normalised version of the mean error. The FGE measures the error in the model, within a range of 0 to 2,
with 0 being ideal, where the model matches the observation.

$$FGE = \frac{2}{n}\sum_i \left| \frac{M_i - O_i}{M_i + O_i} \right| \tag{2}$$

Finally, for correlation, as the data was not normally distributed, the Spearman ranked correlation coefficient was used. The
Spearman ranked correlation coefficient, referred to as just correlation or $\rho$ in this study, is a measure of how linearly correlated
the ranks of the measured and modelled values are. The values range from -1 (negatively correlated) to 1 (positively correlated),
with 1 being perfectly correlated.

$$\rho = \frac{cov(R(O), R(M))}{\sigma_{R(O)}\sigma_{R(M)}} \tag{3}$$

where $R(M)$ is the rank of the modelled values and $R(O)$ is the rank of the observed (experimental) values.

### 4    Intercomparison and evaluation of recent DMS mechanisms

The modified mean bias (MMB), fractional gross error (FGE), and correlation ($\rho$) for the species in all the experiments we
modelled (with all the mechanisms) can be found in the supplementary information (Section S4), however, the outcomes of the
study are summarised in the following figure.



Figure 4 shows how the mechanisms perform for each species. In the case where more than one experiment measured a certain species, the metrics for that species were averaged between the different experiments. For example, as DMSO was

measured by all the experiments, to obtain the average MMB for DMSO for the Shen mechanism, the MMB from the DMSO modelled in each of the experiments (Jernigan et al. (2022); Ye et al. (2022); Shen et al. (2022); Albu et al. (2008)) with the Shen mechanism were averaged. The individual MMBs for DMSO found for each experiment modelled with the Shen mechanism are also shown in Figure 4, with different symbols corresponding to different experiments. This was done for all species and all mechanisms.

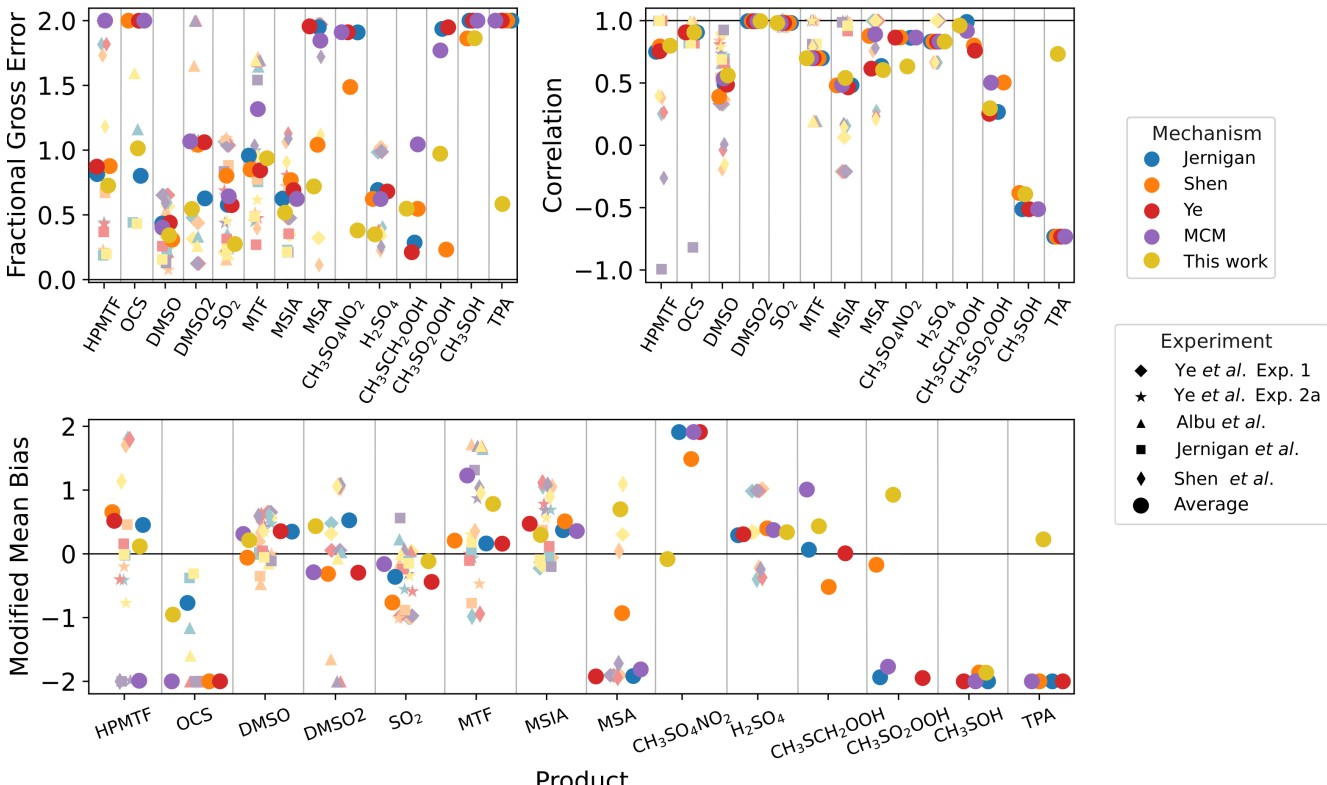

**Figure 4.** The average fractional gross error, correlation, and modified mean bias of the five mechanisms (Jernigan, Shen, Ye, MCM and this work) for each product found in the experiments (Albu et al. (2008); Ye et al. (2022); Jernigan et al. (2022); Shen et al. (2022))

We note that none of the statistical metrics we calculated are perfect descriptors for the performance of a mechanism against experimental data. However, the use of these metrics provides a succinct and quantitative way for the evaluation of the mechanisms to be performed. An idealised mechanism would have an average FGE and MMB of 0 for all compounds, and a $\rho$ of 1. Figure 4 demonstrates that none of the mechanisms are ideal, with the performance of the mechanisms differing over the different compounds. The metrics from the individual experiments (shown as the fainter symbols) also show a large range

between different experiments, even for the same mechanism. This spread is demonstrated with MTF (CH$_3$SCHO), where the



Jernigan mechanism shows a negative bias of around -1 when modelling the Shen et al. (2022) experiment, and a positive bias of 1.6 for the Albu et al. (2008) experiment. Although the average bias of the Jernigan mechanism for MTF of 0.16 is close to the ideal bias of 0, the deviations from the experiments are captured in the larger average error of 0.96. This demonstrates that although the MMB is a useful metric in assessing the performance of a mechanism and provides more information than the FGE, an average bias of close to zero may not be indicative of a mechanism performing well in replicating the production of a compound; the range in the bias from different experiments should be taken into account. However, the average FGE does provide a useful summary of the overall performance of each mechanism.

For 8 of the 14 species included in this study, the mechanism developed in this work had the lowest average FGE. For 12 species, the mechanism from this study is among the two mechanisms with the lowest error. The average FGE for the mechanism was 0.70, with most species having an error lower than 1. The MCM has the highest or equal highest average error for 8 of the species, with four of those species not being formed by the MCM (resulting in an FGE of 2). The high errors from the MCM tended to come from its poor performance for the experiments conducted in low $NO_x$ conditions. These experiments were where HPMTF formation dominates, a pathway missing from the MCM.

The large spread in the error and bias between the mechanisms demonstrates that the adjustments between the mechanisms, sometimes as little as 9 or 12 reactions in the case of the Jernigan and Ye mechanisms, are important in the modelling of these experiments. Since the mechanisms being adjusted and are compared to only one set of experiments, they tend to perform better for their own experiment compared to the others. This could be due to the experimental conditions impacting which reactions are important, or the different products measured in the experiments. One such example is the Shen mechanism having the largest average error for $SO_2$ (0.80). The Shen mechanism underestimates the $SO_2$ in the Jernigan et al. (2022) and Ye et al. (2022) experiments, resulting in an average bias of -0.76. Although $SO_2$ was expected as a major product, it was not measured in the Shen et al. (2022) experiment, and as such the mechanism could not have been evaluated for its performance of modelling $SO_2$. This highlights the need for mechanism development to include a range of mechanisms and experiments.

The Spearman rank correlation coefficient ($\rho$) provides a useful way to assess the correlation between the mechanisms and the experiments, with two caveats. This form of correlation is a measure of whether the observed values and modelled values are both increasing or decreasing during the same time step. However, in one timestep, if the increase/decrease in the observed value due to noise is larger than the actual increase/decrease in the concentration of a compound, this will affect the $\rho$ found. The significant noise in the concentration of some products in the Shen et al. (2022) experiment, the OCS concentration in the Jernigan et al. (2022) experiment, and the $SO_2$ concentration in the Ye et al. (2022) experiment 2a, contribute to lower $\rho$. Additionally, as the Shen et al. (2022) and Jernigan et al. (2022) experiments are steady-state experiments, once the experiment reaches steady-state, small deviations in the experiment can result in lower $\rho$ values. However, the reduction in $\rho$ due to noise and the experiments reaching a steady state will affect the performance of all mechanisms similarly, and the range in correlation found between the mechanisms for each compound is a useful metric in assessing the performance of a mechanism. For 8 of the 14 compounds, the mechanism developed in this work has the highest or equal highest correlation, with the mechanism having the highest or second highest correlation for 10 compounds.



In an ideal world, a developed mechanism could approach the 'perfect' FGE and MMB of 0 and $\rho$ of 1. Deviations from the ideal can be attributed to uncertainties and unknowns in the rate constants and reactions of the mechanism, although that would assume that the experiments themselves represent the 'truth'. In reality, there are uncertainties in the concentrations of the products, especially in the case of the low concentrations measured in the experiments, and the difficulty in determining the sensitivities of the species measured. The compounds measured using chemical ionization mass spectrometers (CIMS), such

as HPMTF, $CH_3SO_4NO_2$, MSIA (in the case of the Jernigan et al. (2022) experiment), and TPA are affected by the largest uncertainties, with Ye et al. (2022) including a 50% relative standard deviation to the species they measured by I-CIMS. In the case of Jernigan et al. (2022), in addition to the large uncertainties in the experimentally determined sensitivity of HPMTF, the TPA and MSIA sensitivities were determined by calculating the species binding energy to iodine, then comparing them to the binding energy of HPMTF, and scaling the experimentally determined HPMTF sensitivity. These uncertainties again emphasise

the importance of comparing multiple experiments from different sources when developing and evaluating a mechanism.

## 5   Discussion of key products

We now focus on a subset of DMS oxidation products. These products (DMSO, HPMTF, MSA and $SO_2$) were chosen as they are found in field studies (Barnes et al., 2006; Veres et al., 2020), and were modelled differently by the mechanisms.

### 5.1   Dimethyl sulfoxide

Firstly, we evaluate the performance of mechanisms in simulating DMSO. DMSO is a primary oxidation product of DMS (formed from both OH-addition and halogen oxide reactions; Barnes et al. (2006)). The modelled DMSO from most mechanisms were similar, with the exception of the Shen mechanism. The Shen mechanism is based on the Hoffmann et al. (2016) mechanism, which uses the explicit mechanism for the OH addition to DMS; the addition of OH to DMS is reversible, forming $CH_3S(OH)CH_3$, which can react with $O_2$ irreversibly to form HODMSO2. These reactions, along with their recommended

rate constants from the 2019 NASA panel report, are included in Table 2.

**Table 2.** The reversible OH addition reaction of DMS, along with the addition of $O_2$, with rate constants from the 2019 NASA panel report

| | Reaction | | | Rate constant |
|---|---|---|---|---|
| 1 | DMS + OH | $\rightarrow$ | $CH_3S(OH)CH_3$ | $3 \times 10^{-31} \times \frac{T}{298}^{-6.24} \times [M]$ |
| 2 | $CH_3S(OH)CH_3 + O_2$ | $\rightarrow$ | HODMSO2 | $8.5 \times 10^{-13}$ |
| 3 | $CH_3S(OH)CH_3$ | $\rightarrow$ | DMS + OH | $\frac{3 \times 10^{-31} \times \frac{T}{298}^{-6.24} \times [M]}{9.6 \times 10^{-27} \times e^{5376/T}}$ |

   The MCM, along with the Jernigan and Ye mechanisms, combine the three reactions from Table 2 into one, using the combined rate constant from IUPAC, $2.2 \times 10^{-12}$ cm$^3$ molecules$^{-1}$ s$^{-1}$ at 298 K and 1 atm (Atkinson et al., 2004). In the Hoffmann et al. (2016) paper, they use the same combined rate constant, but for the forward (reversible) reaction, referencing the MCM. However, this combined rate constant is slower than the forward reaction recommended by the 2019 NASA panel



report ($7.4 \times 10^{-12}$ cm$^3$ molecules$^{-1}$ s$^{-1}$ at 1 atm and 298 K), as the combined reaction takes into account the backward reaction. The rate constant used by Hoffmann et al. (2016) for the backward reaction ($2.3 \times 10^6$ s$^{-1}$ at 298 K) is from Lucas and Prinn (2002), which is slower than the backward reaction from the 2019 NASA panel report ($1.1 \times 10^7$ s$^{-1}$). However, due to the fast reaction of $CH_3S(OH)CH_3$ with $O_2$, the slower forward reaction used by Hoffmann et al. (2016) (and the Shen mechanism) results in less DMSO being produced, which is why less DMSO is formed via the Shen mechanism.

## 5.2 Hydroperoxymethyl thioformate

Recent global modelling (Fung et al., 2022; Cala et al., 2023) point to HPMTF being a major oxidation product of DMS that was unaccounted for until very recently. The major uncertainties surrounding the modelling of HPMTF are the first isomerisation (H-shift of $CH_3SCH_2O_2$), along with the reactions of $CH_3SCH_2O_2$, and uptake of HPMTF onto aerosol surfaces (Cala et al., 2023; Assaf et al., 2023). HPMTF was measured in the Jernigan et al. (2022) experiment, Ye et al. (2022) experiment 2a, and the Shen et al. (2022) experiment; the observed HPMTF, along with the HPMTF modelled by the various mechanisms are shown in Figure 5.

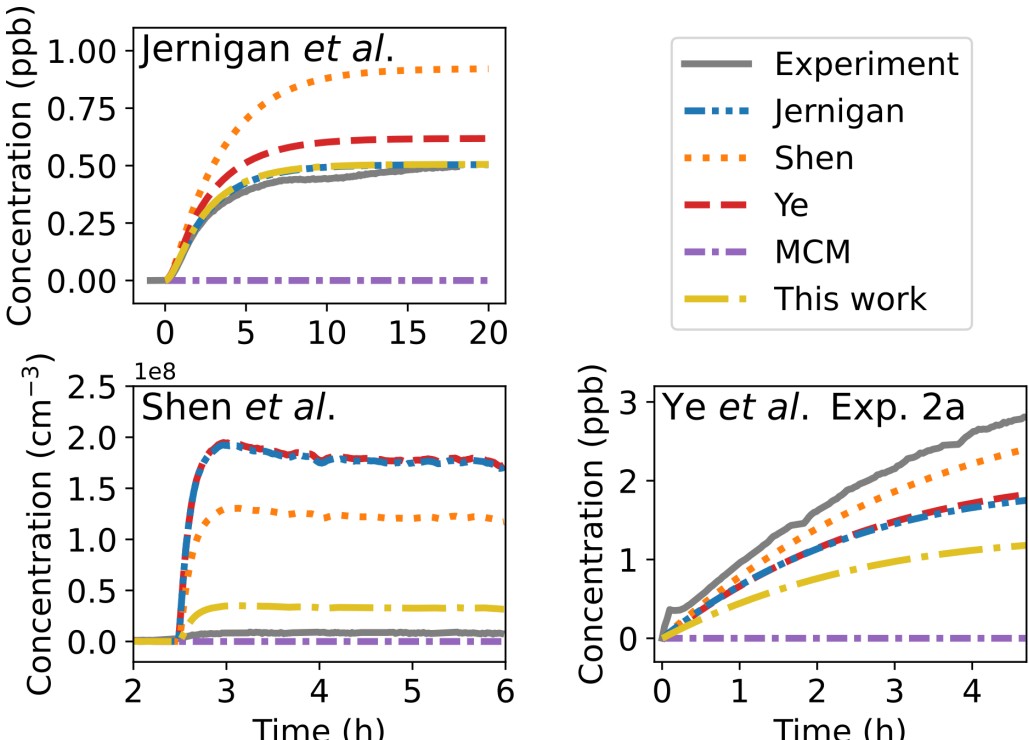

**Figure 5.** Comparison of modelled and measured HPMTF in the Ye et al. (2022) experiment 2a, Jernigan et al. (2022) and Shen et al. (2022) experiments. The measured HPMTF is shown in grey and modelled HPMTF, using the various mechanisms, is shown using the same colours and linestyles as Figure 3.



In the mechanism developed in this work, the Assaf et al. (2023) temperature-dependent rate constant was used for the first H-shift ($0.06$ s$^{-1}$ at 298 K). This rate constant was used as it was both measured directly and is temperature dependent, however, it is slower than the other rate constants measured at 298 K (Berndt et al. (2019); Ye et al. (2021, 2022); Jernigan et al. (2022)). The HPMTF formed is sensitive to this reaction. In the Jernigan et al. (2022) experiment, our new mechanism outputs the second lowest concentration of HPMTF, with 0.2% more HPMTF forming than the Jernigan mechanism. However, the Jernigan mechanism includes a larger rate constant for the RO$_2$ reaction of CH$_3$SCH$_2$O$_2$, which reduces the HPMTF formed for that experiment.

The rate constant used by Jernigan et al. (2022) is the rate constant recommended by the 2019 NASA panel report for the CH$_3$SCH$_2$O$_2$ self reaction ($1.0 \times 10^{-11}$ cm$^3$ molecules$^{-1}$ s$^{-1}$). All the other mechanisms use the rate constant from the MCM v3.3.1 ($3.7 \times 10^{-12}$ cm$^3$ molecules$^{-1}$ s$^{-1}$), which uses the same NASA panel report recommended self-reaction rate constant. Since the MCM uses a pooled RO$_2$ concentration instead of explicitly representing the RO$_2$ reactions, the rate constant they use for RO$_2$ reactions is double the geometric mean of the self-reaction rate constant of the species in question, and the self-reaction rate constant for CH$_3$O$_2$ at 298 K, $3.5 \times 10^{-13}$ cm$^3$ molecules$^{-1}$ s$^{-1}$ (Jenkin et al., 1997). This method is used as CH$_3$O$_2$ is generally the major RO$_2$ reacting; in the case of these experiments, CH$_3$SCH$_2$O$_2$ makes up 1-30% of the RO$_2$ pool, whereas CH$_3$O$_2$ makes up 40-90% of the RO$_2$ pool (based on our modelling results). Jernigan et al. (2022) uses a higher rate constant for the CH$_3$SCH$_2$O$_2$ reaction with pooled RO$_2$, which is why the Jernigan mechanism forms less HPMTF than the other mechanisms in their experiment (due to its shorter lifetime of CH$_3$SCH$_2$O$_2$). As the Jernigan et al. (2022) experiment involved the ozonolysis of TME, it had a higher concentration of total RO$_2$ compared to Ye et al. (2022) experiment 2a and the Shen et al. (2022) experiment by 383% and 563%, respectively. The larger RO$_2$ concentration meant that the different rate constant used for the RO$_2$ reaction of CH$_3$SCH$_2$O$_2$ was more significant for the Jernigan et al. (2022) experiment.

Another source of difference between the mechanisms is the rate constant used for the OH-initiated oxidation of HPMTF. The Shen mechanism uses the slowest rate constant, $1.4 \times 10^{-12}$ cm$^3$ molecules$^{-1}$ s$^{-1}$, which was based off the computational paper by Wu et al. (2015). The rate constant Jernigan et al. (2022) used, $1.4 \times 10^{-11}$ cm$^3$ molecules$^{-1}$ s$^{-1}$, was based on the best fit to their experiment, however, that fit is dependent on the reactions forming HPMTF. Ye et al. (2022) used $1.0 \times 10^{-11}$ cm$^3$ molecules$^{-1}$ s$^{-1}$, which was based on Vermeuel et al. (2020), who found their rate constant through the best fit to observations, and the assumption that the rate constant will be similar to the rate constant measured for MTF (due to structural similarities). The rate constant used in this work, $1.75 \times 10^{-11}$ cm$^3$ molecules$^{-1}$ s$^{-1}$, is an average of the rate constant obtained by Ye et al. (2022) ($2.1 \times 10^{-11}$ cm$^3$ molecules$^{-1}$ s$^{-1}$, found by looking at the decay of HPMTF after adding NO) and the value of best fit from the Jernigan et al. (2022) study.

The importance of the rate constant used for the isomerisation of HOOCH$_2$SCH$_2$O$_2$ into HPMTF is dependent on the other reactions of HOOCH$_2$SCH$_2$O$_2$. Other than isomerisation, the two reactions included in most mechanisms are the reaction with NO and the reaction with HO$_2$. In addition, in our mechanism we included the reactions with RO$_2$ and NO$_3$. For the isomerisation of HOOCH$_2$SCH$_2$O$_2$, both the Shen mechanism and the mechanism from this work use the rate constant calculated by Veres et al. (2020), which at 298 K is a factor of 317 slower than calculated in the Wu et al. (2015) study (used in the Ye mechanism). The calculation by Veres et al. (2020) for the first H-shift agrees well with the measured rate constant by





Assaf et al. (2023) at 298 K, which is why the Veres et al. (2020) calculation for the second H-shift was chosen for this work. However, the slower rate constant used in this work results in the reaction of $HOOCH_2SCH_2O_2$ with $HO_2$ and, to a lesser extent $RO_2$, reducing the amount of HPMTF formed.

The rate constants used in the mechanism developed in this work seem to model the HPMTF from the Jernigan et al. (2022) and Shen et al. (2022) experiments the best. However, apart from the MCM, it has the highest fractional gross error for the Ye et al. (2022) experiment 2a of all the mechanisms studied, although an uncertainty of around 50% was included for the HPMTF measured in that experiment.

### 5.3    Methane sulfonic acid

MSA is measured in two experiments, the Shen et al. (2022) experiment and Ye et al. (2022) experiment 1. In these experiments, the modelled MSA from the mechanism developed in this work came from two different reactions involving $CH_3SO_3$. In the Shen et al. (2022) experiment, nearly all of the MSA that was modelled came from the reaction with $HO_2$, however, in the Ye et al. (2022) experiment 1, MSA came from the reaction of $CH_3SO_3$ with DMS.

Apart from the mechanism developed in this work, for the Ye et al. (2022) experiment 1 the other mechanisms produced a
modified mean bias of around -2 for MSA (meaning almost no MSA formed). The Yin et al. (1990) paper, which involved a review and an evaluation of a DMS mechanism, included a few reactions forming MSA where the $CH_3SO_3$ radical abstracted hydrogen from different species. Yin et al. (1990) mention that type of reaction as key for MSA formation; the MCM already includes the $CH_3SO_3$ and $HO_2$ reaction, which is the source of MSA in the Shen et al. (2022) experiment and discussed below. The estimated rate constants for $CH_3SO_3$ + R-H from Yin et al. (1990) were based on the bond dissociation energy of
the relevant bond between hydrogen and the H-donor. We included all the MSA forming reactions from the Yin et al. (1990) paper, however, for this experiment, the model was only sensitive to the $CH_3SO_3$ reaction with DMS. The rate constant for this reaction was increased by a factor of 2.1 until the sulfuric acid ($H_2SO_4$) to MSA ratio was the same as measured in the experiment (as $H_2SO_4$, measured as sulfate in this experiment, is the other major fate of the $CH_3SO_3$ radical). Not only did the addition of this reaction to the mechanism explain the formation of MSA, but it also accounted for the total amount of DMS
reacted with, which was underestimated by the other mechanisms (Figure 6). However, the formation of MSA is dependent on the formation of $CH_3SO_3$ in the model, which depends on $CH_3SO_2$ reactions. The rate constant of $CH_3SO_2$ decomposition was increased to 6 s$^{-1}$ (at 295 K) in our mechanism, discussed in more detail in the following section.

Figure 6 shows that following these adjustments, the mechanism from this work is able to reproduce the major products (MSA, $SO_2$ and $H_2SO_4$) along with the DMS lost during the experiment. Again, this method of tuning is dependent on the
accuracy of the measurements of the different species and the rate constants of other reactions, and is not necessarily accurate. However, it does indicate that more studies should be conducted on the $CH_3SO_2$ and $CH_3SO_3$ reactions, which include large uncertainties, as the modelled DMS oxidation products are sensitive to them.

Our mechanism overestimates MSA in the Shen et al. (2022) experiment, resulting in a modified mean bias of 1.09. The source for the overestimation is likely due to uncertainties in the DMSO and MSIA reactions with OH at 263 K, along with





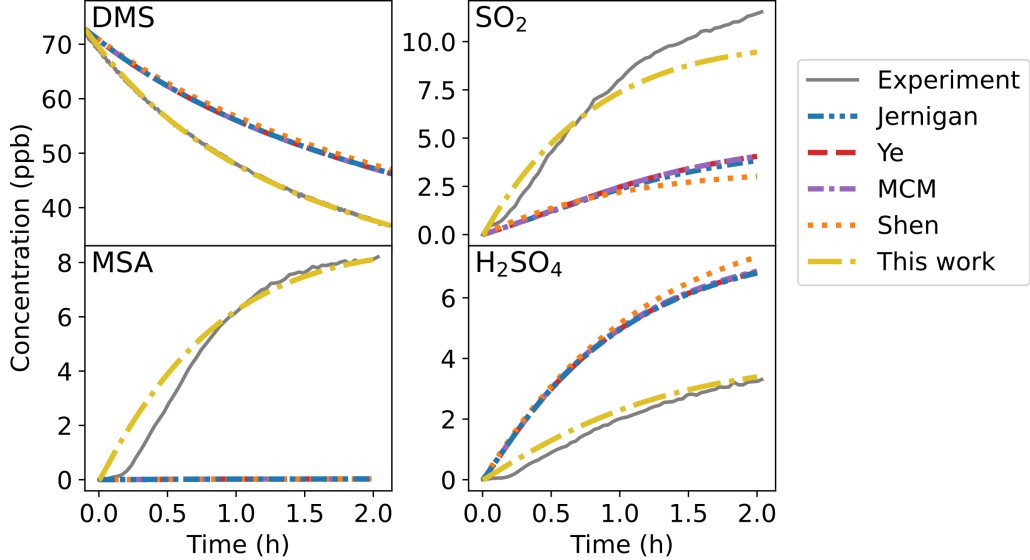

**Figure 6.** The DMS, $SO_2$, MSA and $H_2SO_4$ measured in Ye et al. (2022) experiment 1, along with the modelling outputs from the different mechanisms (Jernigan, Ye, MCM, Shen, and the mechanism developed in this work). The experimental DMS shown has not been corrected for dilution, which was included in the modelled DMS (explained in more detail in supplementary, Section S3).

the reaction of $CH_3SO_3$ and $HO_2$, indicating a need for further temperature dependent experiments. The modelling of MSA for the Shen et al. (2022) experiment is discussed in further detail in the supplementary information (Section S5).

### 5.4   Sulfur dioxide

The modelling of sulfur dioxide ($SO_2$) is complex as it forms from a range of different reactions, which are dependent on the conditions of the experiments. The first column of Figure 7 shows the $SO_2$ measured by the different experiments and

the model output from the mechanisms. The second column in the figure shows the rate of formation of $SO_2$ by the major reactions, from the mechanism developed in this study.

     Our mechanism generally performs similarly to the Ye and Jernigan mechanisms for the total $SO_2$ formed in the experiments modelled. The MCM overestimates the $SO_2$ formed in the Jernigan et al. (2022) experiment, which is due to HPMTF being a major product in this experiment, a product missing in the MCM. The Shen mechanism tends to underestimate $SO_2$ in all

experiments apart from the Albu et al. (2008) experiment. The largest deviations from the other mechanisms are in the low $NO_x$ experiments where HPMTF is a major product; this is mostly due to the Shen mechanism having the slowest rate constant for the reaction of HPMTF and OH radicals, of which $SO_2$ is a secondary product.

     Figure 7 demonstrates that the formation of $SO_2$ from the decomposition of $CH_3SO_2$ is a major reaction for all of the experiments. The Ye and Jernigan mechanisms use the same temperature-dependent rate constant as the MCM (0.4 s$^{-1}$ at 298

K), whereas the Shen mechanism uses a faster temperature-dependent rate constant (7.0 s$^{-1}$ at 298 K) for the decomposition of



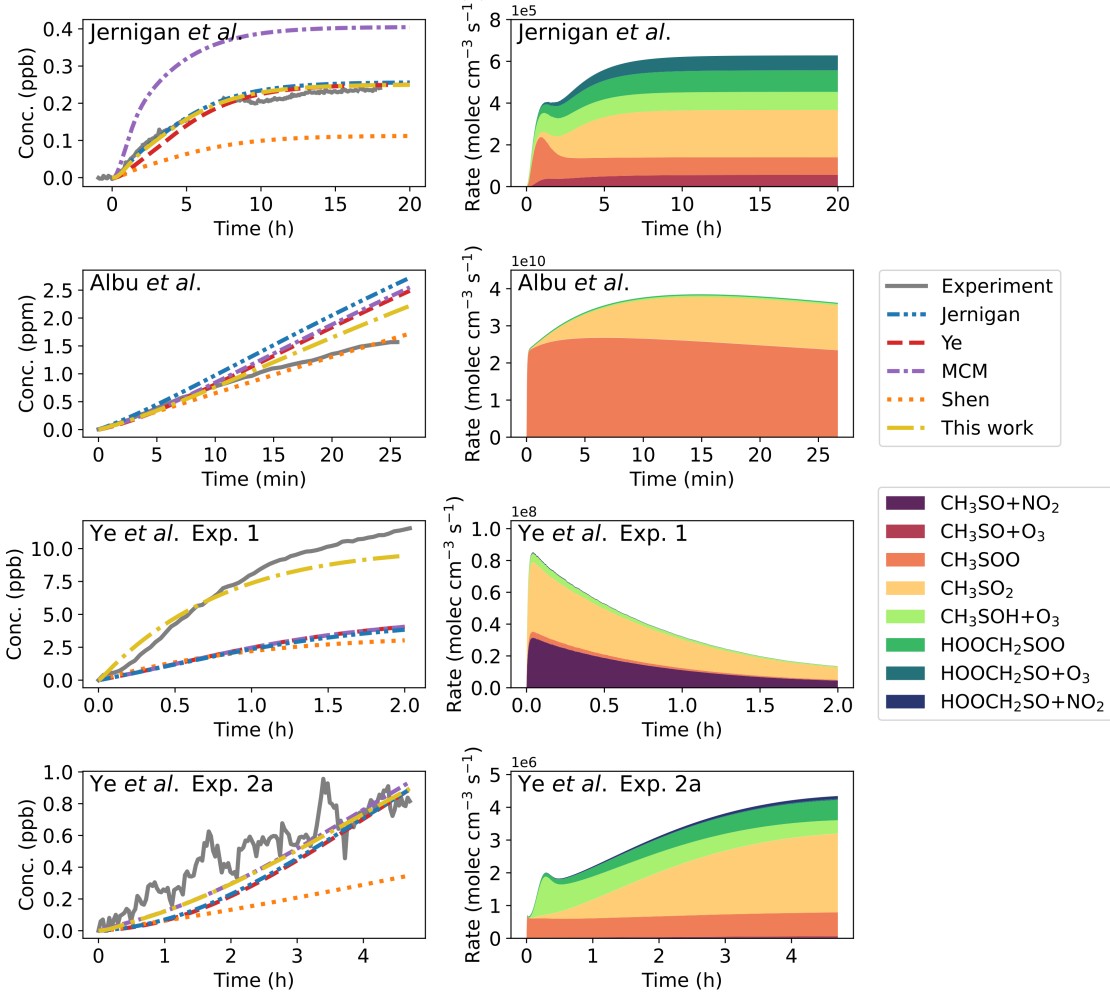

**Figure 7.** The column on the left shows the $SO_2$ formed in the various experiments (Jernigan et al. (2022); Albu et al. (2008); Ye et al. (2022)) compared to the modelled $SO_2$ from the different mechanisms. The column on the right shows the different rates of $SO_2$ formation modelled from our mechanism for those experiments, the reactants of those reactions are included in the legend. Only the major $SO_2$ forming reactions are included.

$CH_3SO_2$. The rate constant used by the MCM is less than the experimental upper bound estimated by Borissenko et al. (2003), $1\,s^{-1}$ at 100-660 Torr and 300 K. Ratliff et al. (2009) determined the barrier for $CH_3SO_2$ dissociation via velocity map imaging to be 14 kcal mol$^{-1}$, and calculated a high-pressure (P = $\infty$) rate constant of $1 \times 10^3\,s^{-1}$. Using a UCCSD(T)//UCCSD level calculation, Chen et al. (2021) also calculated an energy barrier of 14 kcal mol$^{-1}$, however calculated a rate constant of $2 \times 10^3$

$s^{-1}$. More recently, in their box modelling Berndt et al. (2023) used a rate constant of $20\,s^{-1}$ to replicate their experiments. Due to the wide range of rate constants estimated, and the sensitivity of the reaction in the Ye et al. (2022) experiment 1, the



rate constant of the decomposition of $CH_3SO_2$ was adjusted in our mechanism until the formation of MSA (and the loss of DMS) in the model matched the experiment (6 s$^{-1}$ at 295 K).

Although a faster rate constant was used for the decomposition of $CH_3SO_2$ compared to the other mechanisms in the study,
Figure 7 shows that this tuning did not seem to negatively affect the modelling of the experiments. The range of experiments we analysed show the multiple pathways through which $SO_2$ is formed, and that our mechanism performs well in all the conditions studied.

## 6 Atmospheric implications

### 6.1 Marine boundary layer run

We've shown that in simulating the chamber studies, there was a large range in the performance of the mechanisms applied. Although some of the above experiments were performed to simulate realistic, marine conditions, high concentrations of DMS or OH precursors result in different conditions from those found in the marine environment. To compare the mechanisms to marine conditions, and determine the atmospheric implications for the divergence amongst the mechanisms, the remote marine boundary layer box model run by Cala et al. (2023) was used. The same input parameters as Cala et al. (2023) were used, with
the exception of photolysis parameters, where the zenith angle was used to obtain the photolysis rate constants, based on the l, m, and n photolysis parameters used by the MCM (Saunders et al., 2003). The runs were performed over eight cloud-free days, in low $NO_x$ conditions (around 10 ppt), with a zenith angle of zero during solar noon. The planetary boundary layer height (based off Ho et al. (2015)) and temperature were varied throughout the day. The input parameters for the run are included in the supplementary information (Section S6).

The results of the marine boundary layer modelling, shown in Figure 8, demonstrate that there is still a significant spread between the different mechanisms under these more atmospherically relevant conditions. All mechanisms show that $SO_2$ is

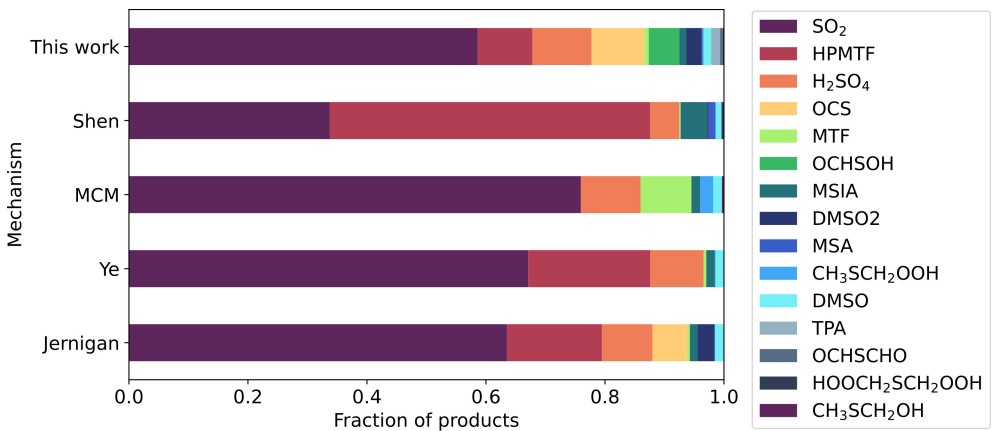

**Figure 8.** The distribution of DMS oxidation products from the marine boundary layer run for each mechanism, based on the average concentration of the products over the last two days of the run.



the major product formed but the range in the fraction of $SO_2$ varies from 0.32 to 0.75 (more than a factor of two). This result is important. Using our new mechanism, which we have demonstrated performs best against the range of experimental chamber studies evaluated, we show that there are much more diverse sets of products formed under atmospheric conditions than most mechanisms would predict. Our calculations imply that the use of the Shen et al. (2022) mechanism would result in

HPMTF being the major gas-phase oxidation product of DMS. Our results are in best agreement with the results of Jernigan et al. (2022) but we note that more detailed observational studies would be required to determine if this wider spread of DMS oxidation products simulated with our mechanism is also seen in reality. We note recent reports of significant amounts of DMSO2 (Edtbauer et al., 2020; Scholz et al., 2023), which the MCM in particular does not predict would form under the conditions investigated but that we calculate would account for approximately 2.6% of DMS oxidation products.

**7   Conclusions**

The oxidation of DMS is complex, but key in understanding the climate impacts of the major natural source of sulfur in the atmosphere. In this work we used the MCM v3.3.1 DMS oxidation scheme as a template to further develop, performed a comprehensive evaluation against an array of recent DMS chamber experiments, and benchmarked the ability of recently proposed DMS mechanisms to simulate this array of experimental data. Basic statistical metrics were applied to determine the

ability of our new mechanism alongside the existing mechanisms to simulate the experimental data. Based on an analysis of these statistics, we concluded that our new mechanism shows greater overall skill in simulating DMS oxidation than the other mechanisms studied. The worst-performing mechanism overall was the MCM, mostly due to the lack of the HPMTF pathway.

However, this work shows that more experimental work needs to be done to reduce the uncertainty in some of the key reactions involved in DMS oxidation. This is especially the case for the rate constants that we adjusted as the model was sensitive

to them, but they hadn't been explored experimentally or computationally; $ROONO_2$ decomposition reactions and direct OCS and DMSO2 formation from the reaction of HPMTF and DMSO with OH, respectively. Additionally, although the decomposition of $CH_3SO_2$ has been experimentally determined previously, there is no consensus in the literature for the decomposition rate constant (Borissenko et al., 2003; Chen et al., 2021; Berndt et al., 2023; Shen et al., 2022), and more experiments should be done to constrain the reaction. Finally, modelling of MSA in the Shen et al. (2022) experiment indicated that further exper-

iments exploring the rate constants for the reactions of DMSO and MSIA with OH radicals at lower temperatures, along with the reaction of $CH_3SO_3$ and $HO_2$, could improve the modelling of MSA in the marine environment.

Additionally, OCS and OCHSOH represent major products of our modelling of marine conditions. These products stem from the OH-initiated oxidation of HPMTF, a pathway that mostly includes structure-activity relationships and theoretical calculations from the Jernigan et al. (2022) paper. As our mechanism is sensitive to these reactions in marine conditions,

further experimental studies should be performed to constrain them.

This paper also highlights the importance of intercomparison studies. By evaluating a mechanism across experiments that include a range of conditions, it reduces the importance of systematic uncertainties in the experiments and ensures the mechanism is robust over a wider range of conditions. Future experiments in different, marine conditions (including reactions with halo-





gens), measuring a wide range of products, would be useful to further constrain the DMS mechanism. To increase the ease of

modelling these experiments in future intercomparison studies, these studies should include the model input files (representing the experimental parameters) in the supplementary details, along with the mechanism files.

*Code and data availability.* The model input files for all the simulated experiments and output files for the model runs using the mechanism developed for this work can be found through Apollo https://doi.org/10.17863/CAM.101652

**Appendix A:  The mechanism developed in this work**

The mechanism developed in this study was based on a thorough literature review, to update and improve the DMS mechanism in the MCM. To determine which rate constants should be used in the model, we used the same construction methodology as the MCM (Saunders et al., 2003). The full mechanism is given in Table A1, with a description of the development of this mechanism included here.

In this methodology, evaluated experimental data took priority. The NASA panel report (Burkholder et al., 2019) and IUPAC

(Atkinson et al., 2004) provide these evaluations on the experimental data, however, the 2019 NASA panel report provides a more recent review, and as such was relied on during this study. In this mechanism, nine of the MCM reactions were updated to the 2019 NASA panel report current recommendations, and 13 reactions were added from the report.

When evaluated experimental data was not provided, published experimental data was used. An additional nine reactions were either adjusted or added with experimental values for rate constants. In the case where there were multiple experiments

that recorded a rate constant (three reactions in this mechanism), the experimental values were either averaged or evaluated to find the rate constant to use. An average that was not used was the $CH_3SCH_2O_2$ H-shift (for the HPMTF pathway). In that case, the rate constant from Assaf et al. (2023) was used due to it being a more direct measurement, and temperature dependent (this decision is discussed in more detail in Section 5.2).

When there were no experimental data to base rate constants on, structure-activity relationships (SARs) or estimates were

used. The MCM provides literature-based SARs related to carbon-based chemistry, however, these SARs do not take into account sulfur chemistry. A comprehensive DMS mechanism paper by Yin et al. (1990) considers sulfur rate experiments, *ab initio* calculations and bond dissociation energies to estimate rate constants that had not been experimentally determined. However, the MCM is based on more recent experiments and includes temperature dependence. To decide between the estimates of Yin et al. (1990) and the SARs of the MCM, we used the following methodology:

– If MCM and Yin et al. (1990) used a similar rate constant at 298 K, the MCM value was used as it includes temperature dependence.

– If MCM and Yin et al. (1990) used different rate constants:

– If Yin et al. (1990) has a sulfur-based reasoning for their rate constant, their value was used



– Otherwise, the MCM value was used, as it is based on more recent literature

If there were no appropriate SAR to use, a theoretical rate constant was used. This was the case for seven reactions. The only theoretical rate constant used that was not for the HPMTF pathway was for the methanesulfinic acid (MSIA) and ozone ($O_3$) reaction from Lv et al. (2019). The other rate constants where theoretical studies were applied were the second H-shift forming HPMTF, and the reactions following the formation of $HOOCH_2S$ from the reaction between HPMTF and OH radicals. Only the major reactions following the HPMTF and OH reaction, based on the theoretical rate constants calculated by Jernigan et al.

(2022), were included in the mechanism. For the second H-shift, we had to choose between two theoretical papers, Veres et al. (2020) and Wu et al. (2015). Veres et al. (2020) note that their rate constants for both H-shifts leading to HPMTF are slower than Wu et al. (2015) (by factors of 51 and 317 at 298 K). Veres et al. (2020) say that this difference is mainly attributed to the different computational methods, and considers their calculations to be more accurate. In addition, the rate constant they calculate for the first H-shift (0.058 s$^{-1}$ at 298 K) agrees well with direct measurements from Assaf et al. (2023), which is why

the Veres et al. (2020) rate constant was used in this work for the second H-shift.

In addition to the above, some reactions were adjusted in this work to better fit the chamber experiments. This was done as these rate constants or branching ratios had not been experimentally determined, however, the model was sensitive to them. Jernigan et al. (2022) found theoretically that carbonyl sulfide (OCS) can form from the decomposition of $HOOCH_2SCO$ (formed from the reaction of HPMTF with OH). However, their predicted branching ratio of direct OCS formation was 3% at

298 K, albeit with an uncertainty factor of at least 3. Although some OCS was formed in our mechanism from the reactions of $HOOCH_2S$, the branching ratio of OCS formed from the initial HPMTF reaction with OH was increased until the modelled OCS matched the OCS observed in the Jernigan et al. (2022) experiment. This branching ratio was found to be 9%, which is within the upper limit of uncertainty calculated by Jernigan et al. (2022)

The decomposition of $ROONO_2$ products formed from our mechanism ($CH_3SCH_2OONO_2$, $CH_3SOO_2NO_2$ and $CH_3SO_4NO_2$,

MSPN) were adjusted to improve the fit to the Ye et al. (2022) experiment 1. The decomposition rate constants for these $ROONO_2$ products in the MCM were not experimentally determined, however, the experiment is sensitive to these reactions. The decomposition rate constant for $CH_3SO_4NO_2$ was increased until the modified mean bias was around 0 when compared to the measured product in the Ye et al. (2022) experiment 1. The same decomposition rate was used for $CH_3SCH_2OONO_2$ (which was not included as a species in the MCM) and $CH_3SOO_2NO_2$. Although these adjustments may not be realistic

and should be adjusted further if they are experimentally determined, due to the low concentration of $NO_x$ in the marine environment, these $ROONO_2$ species are not considered important for modelling DMS in the environment.

One exception to our methodology was the decomposition of $CH_3SO_2$. Although the rate constant of $CH_3SO_2$ was experimentally determined to be less than 1 s$^{-1}$ (Borissenko et al., 2003), both Shen et al. (2022) and Berndt et al. (2023) used a higher rate constant of 7.0 and 20 s$^{-1}$ at 298 K, respectively, to model their experiments. Additionally, Chen et al. (2021)

calculated a decomposition rate constant of $2 \times 10^3$ s$^{-1}$. Due to the large range of rate constants used, and the sensitivity of Ye et al. (2022) experiment 1 to the reaction, in this work the rate constant was set to the best fit. To do this, initially the estimated rate constant of the reaction between $CH_3SO_3$ and DMS from Yin et al. (1990) was increased by a factor of 2.1 for



the sulfuric acid ($H_2SO_4$, measured as sulphate) to methanesulfonic acid (MSA) ratio modelled to be the same as found by Ye et al. (2022) experiment 1 ($1.45 \times 10^{-13}$ cm$^3$ molecules$^{-1}$ s$^{-1}$). Then, the rate constant of the decomposition of $CH_3SO_2$ was

adjusted until the $SO_2$ to MSA ratio modelled by our mechanism was the same as measured by Ye et al. (2022) experiment 1 (6 s$^{-1}$ at 295 K). Both these adjustments are described in more detail in Section 5. These adjustments do not necessarily provide correct rate constants, however, they can be used to improve the fit to the chamber studies until more experiments are performed. Finally, the temperature dependence of $CH_3SO_2$ decomposition (and $CH_3SO_3$) was included by using the same temperature dependence as the MCM.

The following Table A1 lists all the sulfur reactions included in our mechanism, along with the sources of those reactions. Similarly to the MCM, $H_2O$, $O_2$ and $CO_2$ are not included as products or reactants in the mechanism. In the case that $H_2O$ or $O_2$ are reactants, their concentration is included in the rate constant of the reaction (in molecules cm$^{-3}$). The mechanism assumes oxygen is a bath gas and as such the formation of $CH_3$ and $CH_3SCH_2$ radicals are included as $CH_3O_2$ and $CH_3SCH_2O_2$, respectively, due to their fast reaction with oxygen. Additionally, instead of including individual $RO_2$ reactions,

the $RO_2$ radicals formed in the model run are lumped into a total $RO_2$ concentration, which is included in the rate constant calculations of $RO_2$ reactions. The variable M in the rate constants of Table A1 is the total bath gas concentration.

In some cases, acronyms (such as DMSO, DMSO2, MSA) are included instead of the chemical formula. These acronyms are given alongside their structural formula in Figure 1. The exceptions to this are: SA ($H_2SO_4$, sulfuric acid), DMSO2O2 ($CH_3SO_2CH_2O_2$), DMSO2OOH ($CH_3SO_2CH_2OOH$), DMSO2OH ($CH_3SO_2CH_2OH$) and DMSO2O ($CH_3SO_2CH_2O$).

The photolysis rate constants included in Table 1A, J("MTF") and J("CH3OOH"), are wavelength-dependent and calculated for each experiment. In the case of J("MTF"), the absorption cross-section of methyl thioformate ($CH_3SCHO$, MTF) measured by Patroescu et al. (1996) was used, with the quantum yield from the MCM (based off $C_3H_7CHO$). The photolysis rate constant J("CH3OOH") was based on the absorption cross-section and quantum yield used by the MCM.



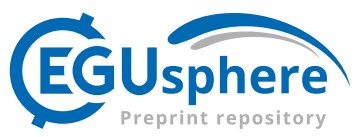

Table A1: The sulfur reactions used in the mechanism developed in this work, along with their source (if not from the MCM) and reference

| | Reaction | | Products | Rate constant | Comment | Ref |
|---|---|---|---|---|---|---|
| 1 | $SO_2 + O$ | = | $SO_3$ | $(\text{KinfT} \times \text{K0T} \times [M]) \times (0.6^{\text{Kb}}) / (\text{KinfT} + \text{K0T} \times [M])$ | Evaluated data | a |
| 2 | $SO_2 + OH$ | = | $HSO_3$ | $(\text{KinfT2} \times \text{K0T2} \times [M]) \times (0.6^{\text{Kb2}}) / (\text{KinfT2} + \text{K0T2} \times [M])$ | Evaluated data | a |
| 3 | $HSO_3$ | = | $HO_2 + SO_3$ | $1.3 \times 10^{-12} \times e^{-330/T} \times [O_2]$ | MCM | b |
| 4 | $SO_3$ | = | $SA$ | $8.5 \times 10^{-41} \times e^{6540/T} \times [H_2O]^2$ | Evaluated data | a |
| 5 | $SO_3 + NO_2$ | = | $NSO_5$ | $1.00 \times 10^{-19}$ | Evaluated data | a |
| 6 | $DMS + NO_3$ | = | $CH_3SCH_2O_2 + HNO_3$ | $1.9 \times 10^{-13} \times e^{530/T}$ | Evaluated data | a |
| 7 | $DMS + OH$ | = | $CH_3SCH_2O_2$ | $1.2 \times 10^{-11} \times e^{-280/T}$ | Evaluated data | a |
| 8 | $DMS + O$ | = | $CH_3SO + CH_3O_2$ | $1.3 \times 10^{-11} \times e^{410/T}$ | Evaluated data | a |
| 9 | $DMS + OH$ | = | $CH_3S(OH)CH_3$ | $[M] \times 3 \times 10^{-31} \times (T/298)^{-6.24} \times [O_2]$ | Evaluated data | a |
| 10 | $CH_3S(OH)CH_3$ | = | $HODMSO2$ | $8.5 \times 10^{-13} \times [O_2]$ | Evaluated data | a |
| 11 | $CH_3S(OH)CH_3$ | = | $CH_3SOH + CH_3O_2$ | $5.0 \times 10^5$ | Estimate | c |
| 12 | $CH_3S(OH)CH_3$ | = | $DMS + OH$ | $[M] \times 3 \times 10^{-31} \times (T/298)^{-6.24} / (9.6 \times 10^{-27} \times e^{5376/T})$ | Evaluated data | a |
| 13 | $CH_3SOH + OH$ | = | $CH_3SO$ | $5.00 \times 10^{-11}$ | Estimate | d |
| 14 | $CH_3SOH + O_3$ | = | $CH_3O_2 + HO_2 + SO_2$ | $2.00 \times 10^{-12}$ | Fit to data | e |
| 15 | $CH_3SCH_2O_2 + HO_2$ | = | $CH_3SCH_2OOH$ | $\text{KRO2HO2} \times 0.387$ | MCM | b |
| 16 | $CH_3SCH_2O_2 + NO$ | = | $CH_3SCH_2O + NO_2$ | $4.9 \times 10^{-12} \times e^{260/T}$ | MCM | b |
| 17 | $CH_3SCH_2O_2 + NO_2$ | = | $CH_3SCH_2OONO_2$ | $9.20 \times 10^{-12}$ | Experiment | f |
| 18 | $CH_3SCH_2O_2 + NO_3$ | = | $CH_3SCH_2O + NO_2$ | $\text{KRO2NO3}$ | MCM | b |
| 19 | $CH_3SCH_2O_2$ | = | $CH_3SCH_2O$ | $3.74 \times 10^{-12} \times [RO_2] \times 0.8$ | MCM | b |
| 20 | $CH_3SCH_2O_2$ | = | $CH_3SCH_2OH$ | $3.74 \times 10^{-12} \times [RO_2] \times 0.1$ | MCM | b |
| 21 | $CH_3SCH_2O_2$ | = | $CH_3SCHO$ | $3.74 \times 10^{-12} \times [RO_2] \times 0.1$ | MCM | b |
| 22 | $CH_3SCH_2O_2$ | = | $HOOCH_2SCH_2O_2$ | $2.39 \times 10^9 \times e^{-7278/T}$ | Experiment | g |
| 23 | $CH_3SCH_2OONO_2$ | = | $CH_3SCH_2O_2 + NO_2$ | $0.134$ | SAR ($CH_3SO_4NO_2$) | h |



| # | Reaction | | Rate | Method | Ref |
|---|---|---|---|---|---|
| 24 | HODMSO2 + NO | = DMSO2 + HO$_2$ + NO$_2$ | $5.0 \times 10^{-12}$ | SAR | c |
| 25 | HODMSO2 | = DMSO + HO$_2$ | $8.90 \times 10^{10} \times e^{-6040/T}$ | MCM | b |
| 26 | CH$_3$SCH$_2$OOH + OH | = CH$_3$SCHO + OH | $7.03 \times 10^{-11}$ | MCM | b |
| 27 | CH$_3$SCH$_2$OOH | = CH$_3$SCH$_2$O + OH | J("CH3OOH") | MCM | b |
| 28 | CH$_3$SCH$_2$O | = CH$_3$S + HCHO | KDEC | MCM | b |
| 29 | CH$_3$SCH$_2$OH + OH | = CH$_3$SCHO + HO$_2$ | $2.78 \times 10^{-11}$ | MCM | b |
| 30 | CH$_3$SCHO | = CH$_3$S + CO + HO$_2$ | J("MTF") | Experiment/SAR | b,i |
| 31 | CH$_3$SCHO + OH | = CH$_3$S + CO | $1.23 \times 10^{-11}$ | Experiment | i,j |
| 32 | DMSO2 + OH | = DMSO2O2 | $1.0 \times 10^{-14}$ | Estimate | c |
| 33 | DMSO + OH | = MSIA + CH$_3$O$_2$ | $6.1 \times 10^{-12} \times e^{800/T} \times 0.9$ | MCM | b,k |
| 34 | DMSO + OH | = DMSO2 + HO$_2$ | $6.1 \times 10^{-12} \times e^{800/T} \times 0.1$ | Estimate | k |
| 35 | DMSO + O | = SO$_2$ + CH$_3$O$_2$ + CH$_3$O$_2$ | $2.0 \times 10^{-12} \times e^{440/T}$ | Evaluated data | a |
| 36 | DMSO + NO$_3$ | = DMSO2 + NO$_2$ | $2.90 \times 10^{-13}$ | Evaluated data | a |
| 37 | CH$_3$S + NO$_2$ | = CH$_3$SO + NO | $3 \times 10^{-11} \times e^{240/T}$ | Evaluated data | a |
| 38 | CH$_3$S + O$_3$ | = CH$_3$SO | $1.5 \times 10^{-12} \times e^{360/T}$ | Evaluated data | a |
| 39 | CH$_3$S | = CH$_3$SOO | $1.20 \times 10^{-16} \times e^{1580/T} \times [O_2]$ | MCM | b |
| 40 | DMSO2O2 + HO$_2$ | = DMSO2OOH | KRO2HO2 $\times 0.387$ | MCM | b |
| 41 | DMSO2O2 + NO | = DMSO2O + NO$_2$ | KRO2NO | MCM | b |
| 42 | DMSO2O2 + NO$_3$ | = DMSO2O + NO$_2$ | KRO2NO3 | MCM | b |
| 43 | DMSO2O2 | = CH$_3$SO$_2$CHO | $2.00 \times 10^{-12} \times [RO_2] \times 0.2$ | MCM | b |
| 44 | DMSO2O2 | = DMSO2O | $2.00 \times 10^{-12} \times [RO_2] \times 0.6$ | MCM | b |
| 45 | DMSO2O2 | = DMSO2OH | $2.00 \times 10^{-12} \times [RO_2] \times 0.2$ | MCM | b |
| 46 | MSIA + OH | = CH$_3$SO$_2$ | $9.00 \times 10^{-11}$ | Evaluated data | a |
| 47 | MSIA + NO$_3$ | = CH$_3$SO$_2$ + HNO$_3$ | $1.0 \times 10^{-13}$ | Estimate | c |
| 48 | MSIA + O$_3$ | = MSA | $1.79 \times 10^{-22}$ | Theory | l |
| 49 | CH$_3$SO + NO$_2$ | = CH$_3$O$_2$ + SO$_2$ + NO | $1.20 \times 10^{-11} \times 0.25$ | MCM | b |
| 50 | CH$_3$SO + NO$_2$ | = CH$_3$SO$_2$ + NO | $1.20 \times 10^{-11} \times 0.75$ | MCM | b |
| 51 | CH$_3$SO + O$_3$ | = CH$_3$O$_2$ + SO$_2$ | $4.00 \times 10^{-13}$ | MCM | b |
| 52 | CH$_3$SO | = CH$_3$SOO$_2$ | $3.12 \times 10^{-16} \times e^{1580/T} \times [O_2]$ | MCM | b |
| 53 | CH$_3$SOO + NO | = CH$_3$SO + NO$_2$ | $1.10 \times 10^{-11}$ | MCM | b |



| # | Reactants | | Products | Rate | Method | Ref |
|---|---|---|---|---|---|---|
| 54 | $CH_3SOO + NO_2$ | = | $CH_3SO + NO_3$ | $2.20 \times 10^{-11}$ | MCM | b |
| 55 | $CH_3SOO$ | = | $CH_3O_2 + SO_2$ | $5$ | Fit to data | m |
| 56 | $CH_3SOO$ | = | $CH_3S$ | $1.5 \times 10^5$ | Fit to data | m |
| 57 | $CH_3SOO + HO_2$ | = | $CH_3SOOH$ | $4.0 \times 10^{-12}$ | SAR | c |
| 58 | $CH_3SOOH + OH$ | = | $CH_3SOO$ | $3.68 \times 10^{-13} \times e^{635/T}$ | SAR (ROOH) | n |
| 59 | $CH_3SOOH$ | = | $CH_3SO + OH$ | J("CH3OOH") | SAR (CH$_3$OOH) | b |
| 60 | $DMSO2OOH + OH$ | = | $CH_3SO_2CHO + OH$ | $1.26 \times 10^{-12}$ | MCM | b |
| 61 | $DMSO2OOH + OH$ | = | $DMSO2O2$ | $3.60 \times 10^{-12}$ | MCM | b |
| 62 | $DMSO2OOH$ | = | $DMSO2O + OH$ | J("CH3OOH") | MCM | b |
| 63 | $DMSO2O$ | = | $CH_3SO_2 + HCHO$ | KDEC | MCM | b |
| 64 | $CH_3SO_2CHO + OH$ | = | $CH_3SO_2 + CO$ | $1.78 \times 10^{-12}$ | MCM | b |
| 65 | $CH_3SO_2CHO$ | = | $CH_3SO_2 + CO + HO_2$ | J("MTF") | SAR (MTF) | b, i |
| 66 | $DMSO2OH + OH$ | = | $CH_3SO_2CHO + HO_2$ | $5.23 \times 10^{-13}$ | MCM | b |
| 67 | $DMSO2OH + OH$ | = | $DMSO2O$ | $1.40 \times 10^{-13}$ | MCM | b |
| 68 | $CH_3SO_2 + O_3$ | = | $CH_3SO_3$ | $3.00 \times 10^{-13}$ | MCM | b |
| 69 | $CH_3SO_2$ | = | $CH_3O_2 + SO_2$ | $1.04 \times 10^{15} \times e^{-9673/T}$ | Fit to data | b,h |
| 70 | $CH_3SO_2$ | = | $CH_3SO_2O_2$ | $1.03 \times 10^{-16} \times e^{1580/T} \times [O_2]$ | MCM | b |
| 71 | $CH_3SO_2 + NO_2$ | = | $CH_3SO_3 + NO$ | $2.20 \times 10^{-12}$ | Evaluated data | a |
| 72 | $CH_3SO_2 + OH$ | = | $MSA$ | $5.0 \times 10^{-11}$ | Estimate | c |
| 73 | $CH_3SOO_2 + HO_2$ | = | $CH_3SO_2 + OH$ | KAPHO2 ×0.44 | MCM | b |
| 74 | $CH_3SOO_2 + HO_2$ | = | $CH_3SOOOH$ | KAPHO2 ×0.41 | MCM | b |
| 75 | $CH_3SOO_2 + HO_2$ | = | $MSIA + O_3$ | KAPHO2 ×0.15 | MCM | b |
| 76 | $CH_3SOO_2 + NO$ | = | $CH_3SO_2 + NO_2$ | $1.00 \times 10^{-11}$ | MCM | b |
| 77 | $CH_3SOO_2 + NO_2$ | = | $CH_3SOO_2NO_2$ | $1.20 \times 10^{-12} \times (T/300)^{-0.9}$ | MCM | b |
| 78 | $CH_3SOO_2 + NO_3$ | = | $CH_3SO_2 + NO_2$ | KRO2NO3 ×1.74 | MCM | b |
| 79 | $CH_3SOO_2$ | = | $CH_3SO$ | $9.10 \times 10^{10} \times e^{-3560/T}$ | MCM | b |
| 80 | $CH_3SOO_2$ | = | $CH_3SO_2$ | $1.00 \times 10^{-11} \times [RO_2] \times 0.7$ | MCM | b |
| 81 | $CH_3SOO_2$ | = | $MSIA$ | $1.00 \times 10^{-11} \times [RO_2] \times 0.3$ | MCM | b |
| 82 | $CH_3SO_3 + HO_2$ | = | $MSA$ | $5.00 \times 10^{-11}$ | MCM | b |
| 83 | $CH_3SO_3$ | = | $CH_3O_2 + SO_3$ | $3.34 \times 10^{13} \times e^{-9946/T}$ | Experiment | b,o |





| # | Reactants | | Products | Rate | Source | Note |
|---|---|---|---|---|---|---|
| 84 | $CH_3SO_3$ + MSIA | = | MSA + $CH_3SO_2$ | $1.0 \times 10^{-13}$ | Estimate | c |
| 85 | $CH_3SO_3$ + HCHO | = | MSA + $HO_2$ + CO | $1.6 \times 10^{-15}$ | Estimate | c |
| 86 | $CH_3SO_3$ + HONO | = | MSA + $NO_2$ | $6.6 \times 10^{-16}$ | Estimate | c |
| 87 | $CH_3SO_3$ + $H_2O_2$ | = | MSA + $HO_2$ | $3.0 \times 10^{-16}$ | Estimate | c |
| 88 | $CH_3SO_3$ + $CH_3OOH$ | = | MSA + $CH_3O_2$ | $3.0 \times 10^{-16}$ | Estimate | c |
| 89 | $CH_3SO_3$ + $CH_3OH$ | = | MSA + $HO_2$ + HCHO | $1.0 \times 10^{-16}$ | Estimate | c |
| 90 | $CH_3SO_3$ + DMS | = | MSA + $CH_3SCH_2O_2$ | $1.45 \times 10^{-13}$ | Estimate | c,h |
| 91 | $CH_3SO_2O_2$ + $HO_2$ | = | $CH_3SO_2OOH$ | KAPHO2 ×0.41 | MCM | b |
| 92 | $CH_3SO_2O_2$ + $HO_2$ | = | $CH_3SO_3$ + OH | KAPHO2 ×0.44 | MCM | b |
| 93 | $CH_3SO_2O_2$ + $HO_2$ | = | MSA + $O_3$ | KAPHO2 ×0.15 | MCM | b |
| 94 | $CH_3SO_2O_2$ + NO | = | $CH_3SO_3$ + $NO_2$ | $1.00 \times 10^{-11}$ | MCM | b |
| 95 | $CH_3SO_2O_2$ + $NO_2$ | = | $CH_3SO_4NO_2$ | $1.20 \times 10^{-12} \times (T/300)^{-0.9}$ | MCM | b |
| 96 | $CH_3SO_2O_2$ + $NO_3$ | = | $CH_3SO_3$ + $NO_2$ | KRO2NO3 ×1.74 | MCM | b |
| 97 | $CH_3SO_2O_2$ | = | $CH_3SO_2$ | $3.01 \times 10^{10} \times e^{-3560/T}$ | MCM | b |
| 98 | $CH_3SO_2O_2$ | = | $CH_3SO_3$ | $1.00 \times 10^{-11} \times [RO_2] \times 0.7$ | MCM | b |
| 99 | $CH_3SO_2O_2$ | = | MSA | $1.00 \times 10^{-11} \times [RO_2] \times 0.3$ | MCM | b |
| 100 | $CH_3SOOOH$ + OH | = | $CH_3SOO_2$ | $9.00 \times 10^{-11}$ | MCM | b |
| 101 | $CH_3SOOOH$ | = | $CH_3SO_2$ + OH | J("CH3OOH") | MCM | b |
| 102 | $CH_3SOO_2NO_2$ + OH | = | MSIA + $NO_2$ | $1.00 \times 10^{-11}$ | MCM | b |
| 103 | $CH_3SOO_2NO_2$ | = | $CH_3SOO_2$ + $NO_2$ | 0.134 | SAR ($CH_3SO_4NO_2$) | h |
| 104 | MSA + OH | = | $CH_3SO_3$ | $2.24 \times 10^{-14}$ | MCM | b |
| 105 | $CH_3SO_2OOH$ + OH | = | $CH_3SO_2O_2$ | $3.60 \times 10^{-12}$ | MCM | b |
| 106 | $CH_3SO_2OOH$ | = | $CH_3SO_3$ + OH | J("CH3OOH") | MCM | b |
| 107 | $CH_3SO_4NO_2$ + OH | = | $CH_3SO_2O_2$ + $HNO_3$ | $3.60 \times 10^{-13}$ | MCM | b |
| 108 | $CH_3SO_4NO_2$ | = | $CH_3SO_2O_2$ + $NO_2$ | 0.134 | Fit to experiment | h |
| 109 | $HOOCH_2SCH_2O_2$ | = | HPMTF + OH | $6.1 \times 10^{11} \times e^{-9.5 \times 10^3/T + 1.1 \times 10^8/T^3}$ | Theory | p |
| 110 | $HOOCH_2SCH_2O_2$ + NO | = | $HOOCH_2SCH_2O$ + $NO_2$ | $4.9 \times 10^{-12} \times e^{260/T}$ | SAR ($CH_3SCH_2O_2$) | b |
| 111 | $HOOCH_2SCH_2O_2$ + $HO_2$ | = | $HOOCH_2SCH_2OOH$ | KRO2HO2 ×0.387 | SAR ($CH_3SCH_2O_2$) | b |
| 112 | $HOOCH_2SCH_2O_2$ + $NO_3$ | = | $HOOCH_2SCH_2O$ + $NO_2$ | KRO2NO3 | SAR ($CH_3SCH_2O_2$) | b |
| 113 | $HOOCH_2SCH_2O_2$ | = | $HOOCH_2SCH_2O$ | $3.74 \times 10^{-12} \times [RO_2] \times 0.8$ | SAR ($CH_3SCH_2O_2$) | b |



| 114 | HOOCH$_2$SCH$_2$O$_2$ | = | HOOCH$_2$SCH$_2$OH | $3.74 \times 10^{-12} \times [RO_2] \times 0.1$ | SAR (CH$_3$SCH$_2$O$_2$) | b |
| 115 | HOOCH$_2$SCH$_2$O$_2$ | = | HOOCH$_2$SCHO | $3.74 \times 10^{-12} \times [RO_2] \times 0.1$ | SAR (CH$_3$SCH$_2$O$_2$) | b |
| 116 | HOOCH$_2$SCH$_2$O | = | HOOCH$_2$S + HCHO | KDEC | SAR (CH$_3$SCH$_2$O) | b |
| 117 | HPMTF + OH | = | HOOCH$_2$S + CO | $1.75 \times 10^{-11} \times 0.91$ | Experiment | h, j, k |
| 118 | HPMTF + OH | = | OH + HCHO + OCS | $1.75 \times 10^{-11} \times 0.09$ | Fit to data | h, j, k |
| 119 | HPMTF | = | HOOCH$_2$S + HO$_2$ + CO | J("MTF") | SAR (MTF) | b, i |
| 120 | HPMTF | = | OCHSCHO + OH + HO$_2$ | J("CH3OOH") | SAR (CH$_3$OOH) | b |
| 121 | OCHSCHO + OH | = | OCS + CO + HO$_2$ | $2.6 \times 10^{-11}$ | Theory | k |
| 122 | OCHSCHO | = | OCS + HO$_2$ + CO + HO$_2$ | J("MTF") | SAR (MTF) | b, i |
| 123 | HOOCH$_2$SCH$_2$OH + OH | = | HOOCH$_2$SCHO + HO$_2$ | $2.78 \times 10^{-11}$ | SAR (CH$_3$SCH$_2$OH) | b |
| 124 | HOOCH$_2$SCH$_2$OOH + OH | = | HOOCH$_2$SCH$_2$O$_2$ | $2 \times 3.68 \times 10^{-13} \times e^{635/T}$ | SAR (ROOH) | n |
| 125 | HOOCH$_2$SCH$_2$OOH | = | HOOCH$_2$SCH$_2$O + OH | J("CH3OOH") | SAR (CH$_3$OOH) | b |
| 126 | OCS + O | = | CO + SO | $2.1 \times 10^{-11} \times e^{-2200/T}$ | Evaluated data | a |
| 127 | OCS + OH | = | SO + OH | $7.2 \times 10^{-14} \times e^{-1070/T}$ | Evaluated data | a |
| 128 | SO | = | SO$_2$ + O | $1.6 \times 10^{-13} \times e^{-2280/T} \times [O_2]$ | Evaluated data | a |
| 129 | SO + O$_3$ | = | SO$_2$ | $3.4 \times 10^{-12} \times e^{-1100/T}$ | Evaluated data | a |
| 130 | SO + NO$_2$ | = | SO$_2$ + NO | $1.40 \times 10^{-11}$ | Evaluated data | a |
| 131 | SO + OH | = | SO$_2$ + HO$_2$ | $2.6 \times 10^{-11} \times e^{330/T}$ | Evaluated data | a |
| 132 | HOOCH$_2$S + O$_3$ | = | HOOCH$_2$SO | $1.5 \times 10^{-12} \times e^{360/T}$ | SAR (CH$_3$S) | a |
| 133 | HOOCH$_2$S + NO$_2$ | = | HOOCH$_2$SO + NO | $3.0 \times 10^{-11} \times e^{240/T}$ | SAR (CH$_3$S) | a |
| 134 | HOOCH$_2$S | = | HOOCH$_2$SOO | $1.20 \times 10^{-16} \times e^{1580/T} \times [O_2]$ | SAR (CH$_3$S) | b |
| 135 | HOOCH$_2$SOO | = | TPA + HO$_2$ | $7.13 \times 10^{-31} \times T^{14.02} \times e^{-2556/T}$ | Theory | k |
| 136 | HOOCH$_2$SOO | = | HOOCH$_2$S | $1.50 \times 10^5$ | SAR (CH$_3$SOO) | m |
| 137 | HOOCH$_2$SOO | = | SO$_2$ + HCHO + OH | 5 | SAR (CH$_3$SOO) | m |
| 138 | TPA + OH | = | OCS + OH | $5 \times 10^{-11} \times 0.14$ | Theory | k |
| 139 | TPA + OH | = | OCHSOH + OH | $5 \times 10^{-11} \times 0.86$ | Theory | k |
| 140 | OCHSOH + OH | = | OCS + OH | $1.40 \times 10^{-12}$ | Theory | k |
| 141 | OCHSOH | = | CO + HO$_2$ + SO + HO$_2$ | J("MTF") | SAR (MTF) | b, i |
| 142 | HOOCH$_2$SO + O$_3$ | = | SO$_2$ + HCHO + OH | $4.00 \times 10^{-13}$ | SAR (CH$_3$SO) | b |
| 143 | HOOCH$_2$SO + NO$_2$ | = | SO$_2$ + HCHO + OH + NO | $1.2 \times 10^{-11}$ | SAR (CH$_3$SO) | b |



**References:** [a] NASA panel report (Burkholder et al., 2019) [b] MCM v3.3.1 (Saunders et al., 2003) [c] Yin et al. (1990) [d] Lucas and Prinn (2002) [e] Berndt et al. (2020) [f] Nielsen et al. (1995) [g] Assaf et al. (2023) [h] This work [i] Patroescu et al. (1996) [j] Ye et al. (2022) [k] Jernigan et al. (2022) [l] Lv et al. (2019) [m] Chen et al. (2021) [n] Jenkin et al. (2018) [o] Berndt et al. (2023) [p] Veres et al. (2020)

**Rate constants:** $\mathrm{KinfT} = 4.1 \times 10^{-14} \times (\mathrm{T}/298)^{1.8}$, $\mathrm{K0T} = 1.8 \times 10^{-33} \times (\mathrm{T}/298)^2$, $\mathrm{Kb} = (1 + (\log_{10}(\mathrm{K0T} \times [\mathrm{M}]/\mathrm{KinfT}))^2)^{-1}$, $\mathrm{KinfT2} = 1.7 \times 10^{-12} \times (\mathrm{T}/298)^{0.2}$, $\mathrm{K0T2} = 2.9 \times 10^{-31} \times (\mathrm{T}/298)^{-4.1}$, $\mathrm{Kb2} = (1 + (\log_{10}(\mathrm{K0T2} \times [\mathrm{M}]/\mathrm{KinfT2}))^2)^{-1}$, $\mathrm{KRO2HO2} = 2.91 \times 10^{-13} \times e^{1300/\mathrm{T}}$, $\mathrm{KRO2NO3} = 2.3 \times 10^{-12}$, $\mathrm{J("CH3OOH")} = $ photolysis rate for $\mathrm{CH_3OOH}$, $\mathrm{KDEC} = 1.0 \times 10^6$, $\mathrm{J("MTF")} = $ photolysis rate for MTF $(\mathrm{CH_3SCHO})$, $\mathrm{KRO2NO} = 2.7 \times 10^{-12} \times e^{360/\mathrm{T}}$, $\mathrm{KAPHO2} = 5.2 \times 10^{-13} \times e^{980/\mathrm{T}}$





*Author contributions.* LSDJ developed the mechanism and ran the box model simulations under the supervision of ATA and CG. LSDJ and ATA wrote the manuscript. All authors were involved in helpful discussions and contributed to the manuscript.

*Competing interests.* The authors have no relevant competing interests to declare.

*Acknowledgements.* The authors would like to thank Qing Ye, Matthew B. Goss, Jesse H. Kroll, Jiali Shen, Xu-Cheng He, Christopher M.
Jernigan and Timothy H. Bertram for giving additional information and data to aid in the modelling of their experiments.





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
