# Peer review of "Extension, Development and Evaluation of the representation of the OH-initiated DMS oxidation mechanism in the MCM v3.3.1 framework"

_EGUsphere, 2023_

## Author Comment (AC1)

We would like to thank the editor and referees for their efforts in the peer-review process of our manuscript. Below we reply to the referee's comments. Black text is used for our response, red text for the referee comments, and blue text is used to identify changes we have added to the manuscript (with  used to denote text we deleted).

**Referee 1:**

This study focuses on collecting the recent advances in the chemistry of DMS with OH from laboratory and chamber studies and building from this an updated chemical model for the OH-initiated DMS oxidation. The updated mechanism is built starting from what available in the MCM adding data from NASA JPL and/or IUPAC when available, published experimental data, SAR and theoretical calculations. For some reactions where no data is available and/or where disagreement between published data is present, are tuned to match the observations. The updated mechanism is compared with experimental results and model results for four different chamber experiments performed at different conditions and is then used to model a selected number of sulfur compound in the marine boundary layer. Here a comparison is performed with the different chemical mechanisms from the chamber studies tested.

Overall the paper is well written and clear and offers a valuable addition to the current knowledge on OH-initiated chemistry of DMS by trying and summarizing together the recent discoveries.I do recommend publication after the following point is considered.

I do though have one main comment that I think should be addressed previous publication. The chemical mechanisms highlighted from the Jernigan et al. (2022) study is, in my opinion, incomplete. The authors mention that there are only five reactions added in the study by Jernigan et al. (2022) which are then used to simulate their experiments in the chamber. This is not correct as a detailed chemical mechanism was developed in the study by Jernigan et al. (2022) by use of theoretical calculations and SAR. Indeed, nine of the reactions from the extended mechanism from the Jernigan et al. (2022) work are then included in the updated mechanism generated within this study. As it is then clear that the authors are aware of this detailed mechanism, why is it not tested for the experiments in Jernigan et al. (2022) study but only five (very tuned) reactions are used? I would be interested to see how the detailed mechanisms would compare with other chamber studies and with the updated new mechanisms developed. At the very least, it should be mentioned that the Jernigan et al. (2022) study has a very extensive mechanism mostly based on theory and SARs, modifying the MCM by dozens of reactions and a reason on why the authors have chosen though to only use the five tuned reactions should be given.

We thank the reviewer for their time to read and comment on this manuscript. As mentioned in the comment, we were aware of the more detailed chemical mechanism that was used by Jernigan et al. (2022), however, we decided to use the simplified (tuned) reactions for two reasons.

Firstly, the simplified (first-generation) mechanism performed better at replicating the measured $SO_2$ in the Jernigan et al. (2022) experiment, which we consider an essential product of DMS oxidation to simulate well, and we were interested in seeing how the first-generation mechanism would perform at simulating the other chamber experiments we studied. Indeed, although it was a rather simplified mechanism, with not many adjustments from the MCM, the Jernigan mechanism used in our paper outperformed most of the other mechanisms.

Secondly, although the theoretical calculations are a major step forward in understanding the reactions following the OH-initiated oxidation of HPMTF, as mentioned in the Jernigan paper, this purely theoretical mechanism (multi-generational model) omitted reactions where no calculations were performed, such as the $HOOCH_2S + O_3$ reaction. The addition of the $HOOCH_2S + O_3$ reaction greatly impacts the results (as described in section S9.1 of the Jernigan et al. (2022) supplementary where the results of the *a priori* mechanism runs, which did include that reaction, was discussed). When that reaction is added, virtually no TPA forms under the experimental conditions of Jernigan et al. (2022) (0.33 ppt of TPA forms, compared to around 50 ppt modelled when that reaction is not included, and 17 ppt measured in the experiment).

However, we understand that it would be of interest to see how the multi-generation model performs at modelling the other experiments, and as such it has been added to the supplementary figures. Additionally, in the main text we have now mentioned the multi-generation model:

'The Jernigan mechanism added only five new reactions, and adjusted four existing reactions. Most of the Jernigan et al. (2022) changes involved adding a simplified HPMTF pathway, along with changes to the methylthiomethyl peroxy ($CH_3SCH_2O_2$, MTMP) radical reaction with $RO_2$, and the DMSO reaction with an OH radical. These changes to the MCM were based on the mechanism file included in the supplementary data of the Jernigan et al. (2022) paper, which differed slightly from the table of reactions given in their supplementary information. **Note that the Jernigan et al. (2022) study also included a mechanism with additional adjustments to the MCM, based on theoretical calculations of the reactions following the OH-initiated oxidation of HPMTF (multi-generational mechanism). As this mechanism was outperformed by the simpler, first generational mechanism, the first generational mechanism was used in this study, and referred to as the 'Jernigan mechanism'. The modelling results of the multi-generational model are included in the supplementary (Figures S3, S5, S7, S9 and S11).'**

Is there a reason why SAR seems to be preferred in comparison to explicit theoretical calculations? For example, for reaction 134 a SAR for $CH_3S$ (a non-oxygenate) is a dubious source to represent $HOOCH_2S$ (an oxygenate) compared to a high-level theoretical calculation (Jernigan et al., 2022) as the SAR would not represent H-bonding or electron-withdrawing effects.

The response to the above comment has been combined with the following specific comment:

Page 19 Line 419. Wouldn't it make more sense to use theoretical calculations before estimates?

The hierarchy of rate constant data, including using SAR (or estimates based on experimental data) over theoretical calculations is based on the framework from the MCM (Jerkin et al., 2003). Whilst theoretical, *ab initio*, calculations have become much more accurate in the last few years than in the early 2000s (when the MCM protocol was designed), calculations for reactions involving the oxidation products of DMS are still very divergent. Wu et al. (2015) calculated a rate constant for the isomerisation of MTMP of 2.9 $s^{-1}$ at 298 K which was around a factor of 50 larger than the theoretical calculation of the same process by Veres et al. (2020), and the rate constant measured experimentally by Assaf et al. (2023). Moreover, for the values for the second H-shift (forming HPMTF), the calculated rate constant ranges from 0.58-175 $s^{-1}$ (Wu et al., 2015; Veres et al., 2020; Jernigan et al., 2022). Finally, for the reaction of HPMTF with an OH radical, the rate constant calculated by Jernigan et al. (2022) (6.80 $\times$ $10^{-12}$ $cm^3$ molecules$^{-1}$ $s^{-1}$) and Wu et al. (2015) (1.4 $\times$ $10^{-12}$ $cm^3$ molecules$^{-1}$ $s^{-1}$) are a factor of 2.6-12.5 smaller than used in our mechanism (1.75 $\times$ $10^{-11}$ $cm^3$ molecules$^{-1}$ $s^{-1}$), which was based on experimental values.

Specific comments:

Page 12 Line 252. I wouldn't say that major uncertainties are there for the isomerization rate of $CH_3SCH_2O_2$. By now there is a direct temperature dependent kinetic study (Assaf et al., 2023) in very good agreement with theory (Jernigan et al., 2022). As pointed out in the text, much less is known about the self reaction of $CH_3SCH_2O_2$ or the heterogenous losses of HPMTF. I would also argue that photolysis of HPMTF is also not known.

The measurements of Assaf et al. (2023) are in reasonable agreement with the theory calculations presented by Jernigan et al. (2022), and in good agreement with Veres et al. (2020) but not with Wu et al. (2015). We still lack a direct calibration method for [HPMTF] – there is no primary standard – and the limited experimental studies that determine $k_{iso}$ vary from 0.09 - 0.23 $s^{-1}$ (at around 298 K, Berndt et al., 2019; Ye et al., 2021,2022; Jernigan et al., 2022; Assaf et al., 2023). We agree that there is little known about the photolysis of HPMTF and we have added a discussion of this in the text:

'The major uncertainties surrounding the modelling of HPMTF are the first isomerisation (H-shift of $CH_3SCH_2O_2$), **photolysis of HPMTF**, along with the reactions of $CH_3SCH_2O_2$, and uptake of HPMTF onto aerosol surfaces'

**'Photolysis reactions remain a major source of uncertainty due to the lack of experimental data. The photolysis rate constants in our mechanism are based on structure-activity relationships of compounds that do not contain sulfur, with the exception of the $CH_3SCHO$ absorption cross-section used for the photolysis of the carbonyl group. Although the Ye et al. (2022) experiments included UV lamps with wavelengths between 300-400 nm, which allowed some evaluation of the photolysis reactions in our mechanism, further experiments exploring the photolysis of DMS oxidation products should be performed.'**

Page 13 Line 279. In the study by Jernigan et al. (2022) also a theoretical value for the rate coefficient of HPMTF with OH for each different channels is available.

This paragraph is talking about the total rate constant for HPMTF + OH used by the other mechanisms, and the experimentally determined rate constants used in our mechanism (which are used in preference to theoretical calculations in our mechanism development protocol). A mention of the theoretical HPMTF + OH channel is talked about in the appendix (page 20 lines 450-457 of the updated manuscript)

Page 13 Last paragraph. I guess also here it could be worth mentioning that also in the study by Jernigan et al. (2022) the H-shift for HOOCH2SCH2O2 was explicitly theoretically calculated. The value is quite faster than what found by Veres et al. (2020), ~ 3 s$^{-1}$ at 298K.

This has now been included:

'For the isomerisation of $HOOCH_2SCH_2O_2$, both the Shen mechanism and the mechanism from this work use the rate constant calculated by Veres et al. (2020), which at 298 K is a factor of 317 slower than calculated in the Wu et al. (2015) study (used in the Ye mechanism) **and a factor of 5.6 smaller than calculated in the Jernigan et al. (2022) study**.'

Figure 8. Although I like this figure and find it useful, I think the addition of a table with the percentage of each species listed for each different mechanisms would give even more information in particular for a more detailed comparison.

A table with data used in Figure 8 (as percentages) has been added to the supplementary (Figure S8). However, we stress that these are not yields but the distribution of DMS products in our marine boundary layer box model run. In the case of $SO_2$, the concentration includes background $SO_2$, which would make it more in line with measurements, but again, not a yield.  Additionally, the concentrations are dependent on the conditions of the specific boundary layer box model run.

**Table S8.** The percentage distribution of the products of DMS oxidation, from the average concentration over two days of the marine boundary layer box model run

| Species | Jernigan | Ye | MCM | Shen | This work |
|---|---|---|---|---|---|
| $SO_2$ | 63.5 | 67.1 | 75.9 | 33.7 | 58.4 |
| HPMTF | 16.0 | 20.5 | 0.0 | 53.8 | 9.2 |
| $H_2SO_4$ | 8.5 | 9.0 | 10.1 | 4.9 | 9.9 |
| OCS | 5.8 | 0.0 | 0.0 | 0.0 | 9.8 |
| $CH_3SCHO$ | 0.5 | 0.5 | 8.5 | 0.2 | 0.7 |
| OCHSOH | 0.0 | 0.0 | 0.0 | 0.0 | 5.0 |
| MSIA | 1.3 | 1.4 | 1.5 | 4.5 | 1.1 |
| DMSO2 | 2.9 | 0.0 | 0.0 | 0.2 | 2.6 |
| $CH_3SCH_2OOH$ | 0.1 | 0.1 | 2.2 | 0.0 | 0.2 |
| DMSO | 1.4 | 1.4 | 1.5 | 1.0 | 1.3 |
| TPA | 0.0 | 0.0 | 0.0 | 0.0 | 1.4 |
| MSA | 0.0 | 0.0 | 0.0 | 1.2 | 0.1 |
| $HOOCH_2SCH_2OOH$ | 0.0 | 0.0 | 0.0 | 0.4 | 0.1 |
| $CH_3SCH_2OH$ | 0.0 | 0.0 | 0.3 | 0.0 | 0.0 |

Page 20 Line 430. Here it is stated that theoretical calculations were used if SAR was not available. This seems in disagreement with the statement from page 19 line 419.

We have changed the text to say 'if there were no appropriate SAR **or estimate** to use,...' for clarity and to better reflect the statement in line 419 (of the original manuscript), and the method procedure

Page 20 Line 437. That is not correct, theoretical calculations for that very H-shift are also available in the study by Jernigan et al. (2022).

This is a good point, and that paragraph has now been modified to include the calculation by Jernigan et al. (2022):

'For the second H-shift, we had to choose between  **three** theoretical papers, Veres et al. (2020)**,** Wu et al. 2015 **and Jernigan et al. (2022).**'

References

Assaf, E., Finewax, Z., Marshall, P., Veres, P. R., Neuman, J. A., and Burkholder, J. B.: Measurement of the Intramolecular Hydrogen-Shift Rate Coefficient for the CH3SCH2OO

Radical between 314 and 433 K, J. Phys. Chem. A, https://doi.org/10.1021/acs.jpca.2c09095, 2023.

Jernigan, C. M., Fite, C. H., Vereecken, L., Berkelhammer, M. B., Rollins, A. W., Rickly, P. S., Novelli, A., Taraborrelli, D., Holmes, C. D., and Bertram, T. H.: Efficient Production of Carbonyl Sulfide in the Low-NOx Oxidation of Dimethyl Sulfide, Geophys Res Lett, 49, e2021GL096838, https://doi.org/10.1029/2021GL096838, 2022.

Veres, P. R., Neuman, J. A., Bertram, T. H., Assaf, E., Wolfe, G. M., Williamson, C. J., Weinzierl, B., Tilmes, S., Thompson, C. R., Thames, A. B., Schroder, J. C., Saiz-Lopez, A., Rollins, A. W., Roberts, J. M., Price, D., Peischl, J., Nault, B. A., Møller, K. H., Miller, D. O., Meinardi, S., Li, Q., Lamarque, J. F., Kupc, A., Kjaergaard, H. G., Kinnison, D., Jimenez, J. L., Jernigan, C. M., Hornbrook, R. S., Hills, A., Dollner, M., Day, D. A., Cuevas, C. A., Campuzano-Jost, P., Burkholder, J., Bui, T. P., Brune, W. H., Brown, S. S., Brock, C. A., Bourgeois, I., Blake, D. R., Apel, E. C., and Ryerson, T. B.: Global airborne sampling reveals a previously unobserved dimethyl sulfide oxidation mechanism in the marine atmosphere, Proc. Natl. Acad. Sci. U. S. A., 117, 4505, 2020.

**Referee 2:**

Jacob and co-workers present a new improved chemical mechanism for DMS oxidation. For the most part, this study is well-designed and well-realized, taking account of much of the existing literature measurements and models in a thorough intercomparison exercise. Given the importance of DMS oxidation to the atmospheric sulphur budget, I expect that this study will be of interest to the atmospheric chemistry community. Therefore, I recommend that it is appropriate for publication in Atmospheric Chemistry and Physics after the authors have considered the following points:

We would like to thank the referee for the helpful comments, and for reading our manuscript. Our responses to the specific points are below.

Line 17: I am not sure what the authors mean, when they say that DMS is the largest natural source of sulphur. There is a certain unappealing circularity to the logic of this statement as it stands.

The sentence has been changed to 'Dimethyl sulfide ($CH_3SCH_3$, DMS)  **emissions are** the largest natural source of sulfur in the atmosphere' for clarity. DMS emissions outweigh all other known sources of naturally produced sulfur into the atmosphere, so we don't think there is any circularity to the logic of the statement. In the future, as anthropogenic emissions of sulfur are more tightly controlled, it is very likely that DMS will be the major source of sulfur into the atmosphere.

Line 29: The halogen oxides do not, I presume, "undergo" hydrogen abstraction, but rather participate in/ initiate hydrogen abstraction.

The sentence has been changed to halogen atoms/oxides 'can  **participate in** H-atom abstraction, halogen-atom addition and O-atom addition to the sulfur atom…'

Line 73: This is admittedly minor, but since the authors have done such a careful job otherwise, I would suggest that they ought to hyphenate "gas-phase" in this sentence.

Done

Figure 1: This is an interesting overview figure that helped me to quickly understand the general comparison between the various treatments of DMS oxidation that are available, so for that, I thank the authors. Despite the complexity of this figure, there are some pieces of information that I would have liked to have seen in addition: 1. Which of the molecules have been measured in lab/ field experiments (perhaps these molecules could be coloured accordingly?). 2. Which of these reactions have been measured/ estimated/ calculated

(perhaps the perimeters of these arrows could be coloured accordingly?). I think doing this would give the reader a sense of the state of the knowledge in this subject.

We appreciate the suggestions to further modify the figure, however, we are weary of overcomplicating the figure, and making it harder to understand. Reviewer 1 agrees this is a useful figure but commented that it's almost a bit too complicated, which we agree with.

We trialled the suggestion to include information on which molecules have been measured in the chamber experiments (by making them bold), however, we found that this made the figure too complicated, and harder to understand. As for including which reactions have been measured/estimated/calculated, we believe that that would also clutter the figure. Readers can refer to Table A1 in the appendix to see which reactions are evaluated data, estimates, SAR, experimental data, fits, or calculations.

Figure 2: I didn't really enjoy the combination of chemical abbreviations and chemical line notation. It asks a lot of the reader to keep the structures of all these abbreviations in mind, and I would advise against their use.

Figure 7: Again, this is a comment that relates to chemical abbreviations and chemical line notation. The authors should make more of an effort with the latter. Where are the radical centres? Where are the unsaturated bonds?

Figure 8: Again, chemical line notation is sub-optimal. OCHSOH is a good example, wouldn't it be better to refer to it as O=CHSOH?

The responses to the above comments have been combined due to their similar nature.

The combination of chemical abbreviations and chemical line notation is consistent with the notation in the master chemical mechanism (MCM) which was the framework for our mechanism. The notation used in our mechanism was used throughout the paper to avoid confusion, and the structures of the major products/intermediates are included in Figure 1 as line drawings to help readers with the abbreviations, with exceptions included in pg 21, lines 494-496. However, when referred to in the text for the first time, the notation including unsaturated bonds has now been included:

'... and O-atom addition to the sulfur atom to form dimethyl sulfoxide ($CH_3S=OCH_3$, DMSO).'

'Based on ab initio calculations, Wu et al. (2015) suggested that this could undergo atmospheric autoxidation and generate hydroperoxymethyl thioformate ($O=CHSCH_2OOH$, HPMTF);'

'Some products, such as thioperformic acid (**S=CHOOH,** TPA), were not included by most mechanisms, while others, such as carbonyl sulfide (OCS), were produced in much lower concentrations by some of the mechanisms than was observed experimentally.'

'Additionally, OCS and OCHSOH **(O=CHSOH)** represent major products of our modelling of marine conditions.'

'In the auxiliary mechanism for the Jernigan et al. (2022) experiment, the tetramethylethylene **($(CH_3)_2C=C(CH_3)_2$,** TME) subset of the MCM was included,'

One exception to the use of chemical abbreviations is methyl thioformate (MTF), where the mechanism uses $CH_3SCHO$, but has been referred to throughout the text as MTF. In that case, the text has now been changed to include both the abbreviation and the line notation:

'This spread is demonstrated with  **methyl thioformate** ($CH_3SCHO$**, MTF**), where the Jernigan mechanism shows a negative bias of around -1 when modelling the..'

'... who found their rate constant through the best fit to observations, and the assumption that the rate constant will be similar to the rate constant measured for  **methyl thioformate ($CH_3SCHO$, MTF)** due to structural similarities.

In the figures of the main text, MTF has been changed to $CH_3SCHO$ for consistency.

Figure 3: It would be useful, where possible to include the chemical structures of the chemicals on each of these panels, so that the reader can quickly grasp which molecule is under consideration.

We trialled adding the chemical structures to Figure 3, and Figure 5 (for consistency), however, we found that doing so cluttered the figure. The structures of the species in these figures are included in the mechanism figure (figure 1), and readers can use this figure if they want to know the chemical structure of the species.

Figure 4: I found that the upper two panels were quite congested compared with the lower panel. Why don't the authors retain the same form factor for each of the panels?

We have taken this suggestion on board, and modified Figure 4 to now consist of 3 equal panels:

[Figure]

Lines 189–217: The terms "useful metric", "useful summary" and "useful way" seem insubstantial to me. Useful in what way exactly?

We have now changed the word 'useful' in those lines to the following:

'This demonstrates that although the MMB  **can be used to assess** the performance of a mechanism and provides more information than the FGE'

'However, the average FGE  **can help summarise** the overall performance of each mechanism'

'The Spearman rank correlation coefficient ($\rho$)  **can be used** to assess the correlation between the mechanisms and the experiments, with two caveats.'

'However, the reduction in $\rho$ due to noise and the experiments reaching a steady state will affect the performance of all mechanisms similarly, and the range in correlation found between the mechanisms for each compound  **can still be used to assess** the performance of a mechanism.'

Line 207: It seems quite obvious that mechanisms that are informed by laboratory and field observations will likely be more reliable. However, it also seems plausible that the inclusion of erroneous measurements could introduce a undesirable bias into the model. With this in

mind, it seems like the best idea to include a wide variety of experimental observations to minimize this effect. This appears to be what the authors have done with this study, but I don't think that you made this point clearly in the manuscript. I would suggest that you comment on this idea.

We agree that comparing to a wide range of experiments is very important in reducing bias in mechanism development, and it was the idea behind this study. The last paragraph of the conclusion talks about this, however, to make this point clearer, we have now emphasised this in line 214, and line 238:

'This highlights the need for mechanism development to include a range of mechanisms and experiments, **as is done in this study**.'

'These uncertainties again emphasise the importance of comparing multiple experiments from different sources when developing and evaluating a mechanism, **such as in this study**.'

Additionally, we have adjusted the abstract to emphasise this point:

Understanding dimethyl sulfide (DMS) oxidation can help us constrain its contribution to Earth's radiative balance. Following the discovery of hydroperoxymethyl thioformate (HPMTF) as a DMS oxidation product, a range of new experimental chamber studies have since improved our knowledge of the oxidation mechanism of DMS and delivered detailed chemical mechanisms. However, these mechanisms have not undergone formal intercomparisons to evaluate their performance.

This study aimed to synthesise the recent experimental studies and develop a new, near-explicit, DMS mechanism, through a thorough literature review, . A simple box model was then used with the mechanism to simulate a series of chamber experiments, and evaluated through comparison with four published mechanisms. Our modelling shows that the mechanism developed in this work outperformed the other mechanisms on average when compared to the experimental data, having the lowest fractional gross error for 8 out of the 14 DMS oxidation products studied. A box model of a marine boundary layer was also run, demonstrating that the deviations in the mechanisms seen when comparing them against chamber data are also prominent under more atmospherically relevant conditions.

Although this work demonstrates the need for further experimental work, the mechanism developed in this work ha**s** been evaluated against a range of experiment**s**, **which validate the mechanism, and reduce the bias from individual experiments. Our mechanism** provides a good basis for a near-explicit DMS oxidation mechanism that would include other initiation reactions (e.g., halogens), and can be used to compare the performance of reduced mechanisms used in global models.

Lines 220–230: The reliability of the experimental data is a key consideration. Would it be possible to provide experimental error bars on some of the time-series in Figure 3 for example? Some of these are going to be highly uncertain, I imagine (e.g. TPA and HPMTF).

Being able to visualise the uncertainties at least in one of the figures (Figure 3.) is a good idea, though it relies on the uncertainties included in the Jernigan et al. (2022) paper. Unfortunately, uncertainties are not consistently included for most products in the Jernigan et al. (2022) paper. In the case of HPMTF and MTF ($CH_3SCHO$), the uncertainty in the sensitivity (ncps/pptv) is included as a percentage. Additionally, for OCS measurement, an uncertainty of 10-12 pmol mol$^{-1}$ is described. Uncertainties for the other species measured are not quantified in the Jernigan et al. (2022) paper. Although large, it would be hard to quantify the uncertainties for TPA and MSIA, which rely on comparing calculated binding enthalpies to HPTMF, and the sensitivity measured for HPMTF. No measurement uncertainty was included for DMSO, or $SO_2$.

Due to the differences in the quantification of the uncertainties, and the lack of uncertainty calculations for some species, we think it is best not to include a visual representation of these uncertainties.

Line 239: many O2 additions are reversible to some extent. Do we know for sure that this O2 addition is irreversible?

The statement of an irreversible reaction of $O_2$ with $CH_3S(OH)CH_3$ is from the NASA panel report Evaluation Number 19 (https://jpldataeval.jpl.nasa.gov/ ) reaction I20 where it says 'The OH + $CH_3SCH_3$ reaction is complex, proceeding by both direct H-abstraction and reversible addition pathways. A recommendation for the direct reaction is given separately in Table 1 (see above). The product of the reversible addition pathway reacts with $O_2$ creating an irreversible path as well.' Their evaluation of the reaction is based on a range of experimental papers, and as such the following statement in our paper is in line with the current literature,

'the Shen mechanism is based on the Hoffmann et al. (2016) mechanism, which uses the explicit mechanism for the OH addition to DMS; the addition of OH to DMS is reversible, forming $CH_3S(OH)CH_3$, which can react with $O_2$ irreversibly to form HODMSO2'

Line 244 (and many other examples): It is incorrect to describe a bimolecular rate constant as being "slower" or "faster" than another. They are perhaps best described as being "smaller" or "larger". I would suggest that the authors carefully look through this manuscript for these and related words in order to assess in each case whether they are referring to these constants or (correctly) referring to the rate of chemical change with time.

This is a good point, and has been adjusted throughout the text:

'However, this combined rate constant is  **smaller** than the forward reaction recommended by the 2019 NASA panel report..'

'The rate constant used by Hoffmann et al. (2016) for the backward reaction (2.3 $\times$ $10^6$ $s^{-1}$ at 298 K) is from Lucas et al. (2002), which is  **smaller** than the backward reaction from the 2019 NASA panel report… '

However, due to the fast reaction of $CH_3S(OH)CH_3$ with $O_2$, the  **smaller** forward reaction **rate constant** used by Hoffmann et al. (2016) (and the Shen mechanism) results in less DMSO being produced, which is why less DMSO is formed via the Shen mechanism.

'This rate constant was used as it was both measured directly and is temperature dependent, however, it is  **smaller** than the other rate constants measured at 298 K… '

'The Shen mechanism uses the  **smallest** rate constant,... '

'... the rate constant calculated by Veres et al. (2020), which at 298 K is a factor of 317  **smaller** than calculated in the Wu et al. (2015) study.. '

'However, the  **smaller** rate constant used in this work results in the reaction of $HOOCH_2SCH_2O_2$ with $HO_2$.. '

'... note that their rate constants for both H-shifts leading to HPMTF are  **smaller** than Wu et al. (2015) (by factors of 51 and 317 at 298 K).'

'... this is mostly due to the Shen mechanism having the  **smallest** rate constant for the reaction of HPMTF and OH radicals.. '

'… whereas the Shen mechanism uses a  **larger** temperature-dependent rate constant…'

'Although a  **larger** rate constant was used… '

Figure 5: I don't understand why the units of concentration vary in the panel related to the experiments of Shen et al. I suggest that it is easier to make a direct comparison between studies when the units are consistent throughout.

This was done to be consistent with the units used in the Shen et al. paper, and the data that was provided from the authors. However it is understandable that it makes it harder to directly compare between different experiments. As such, in Figure 2 the units of concentration are now given as both ppb and molecules $cm^{-3}$ for the Shen et al. (2022) experiment:

[Figure]

As suggested for Figure 5., the concentration for the Shen et al. (2022) experiment has now been converted to ppb.

Line 258 (and other similar examples): where possible, it would be best to include uncertainties in rate constants, such that the reader can assess whether these are well-known quantities (or otherwise).

This is another good point, and uncertainties have been included where possible:

'However, this combined rate constant is smaller than the forward reaction recommended by the 2019 NASA panel report (7.4 **± 3.3** $\times$ $10^{-12}$ $cm^3$ molecules$^{-1}$ s$^{-1}$ at 1 atm and 298 K),...'

'In the mechanism developed in this work, the Assaf et al. (2023) temperature-dependent rate constant was used for the first H-shift (0.06 **± 0.02** s$^{-1}$ at 298 K).'

'The rate constant used by Jernigan et al. (2022) is the rate constant recommended by the 2019 NASA panel report for the $CH_3SCH_2O_2$ self reaction (1.0 **± 0.6** $\times$ $10^{-11}$ $cm^3$ molecules$^{-1}$ s$^{-1}$).'

'…, and the self-reaction rate constant for $CH_3O_2$ at 298 K, 3.5 **± 1.5** $\times$ $10^{-13}$ $cm^3$ molecules$^{-1}$ s$^{-1}$ (Jenkin et al.,1997).'

'The rate constant Jernigan et al. (2022) used, 1.4 **(0.27-2.4)** $\times$ $10^{-11}$ cm$^3$ molecules$^{-1}$ s$^{-1}$, was based on the best fit to their experiment… '

The rate constant used in this work, 1.75 $\times$ $10^{-11}$ cm$^3$ molecules$^{-1}$ s$^{-1}$, is an average of the rate constant obtained by Ye et al. (2022) (2.1 **± 0.1** $\times$ $10^{-11}$ cm$^3$ molecules$^{-1}$ s$^{-1}$, found by looking at the decay of HPMTF after adding NO)... '

General comments on mechanism:

I am surprised that photolysis reactions are not considered to be a key uncertainty in this mechanism. When inspecting your Figure 1, I was left unconvinced by your treatment of HOOCH2SCH2OOH and HPMTF. Firstly, why would you expect that HOOCH2SCH2OOH breaks across a C–S bond to yield the S-centred radical SCH2OOH? Isn't it more likely that it breaks across one of the peroxidic bonds to yield an alkoxy that goes towards HPMTF? Secondly, it is possible that its carbonyl moiety will be more photolabile than the hydroperoxide functionality. This would serve to increase the amount of SCH2OOH that you are forming in your model. Thirdly, what about those molecules that possess carbonyl and hydroperoxide groups, but possess no photolysis reactions in your model (e.g. TPA and O=CHSOH)?

In terms of the photolysis reactions unfortunately there is not a lot of experimental data to rely on. In this study, for R-SCHO compounds, the absorption cross-section of MTF (CH$_3$SCHO) measured by Patroescu et al. (1996) was used instead of the absorption cross-section of C$_3$H$_7$CHO, which is currently used by the MCM for these Norrish type I (carbonyl) photolysis reactions. This was done to reduce the uncertainty in the photolysis reactions. However for quantum yield, the yields of C$_3$H$_7$CHO were used (as with the MCM), due to lack of experimental data (this is discussed briefly in lines 497-500). A paragraph in the conclusion has now been added to emphasise the uncertainty in the mechanism due to photolysis reactions:

**'Photolysis reactions remain a major source of uncertainty due to the lack of experimental data. The photolysis rate constants in our mechanism are based on structure-activity relationships of compounds that do not contain sulfur, with the exception of the CH$_3$SCHO absorption cross-section used for the photolysis of the carbonyl group. Although the Ye et al. (2022) experiments did include UV lamps with wavelengths between 300-400 nm, which allowed some evaluation of the photolysis reactions in our mechanism, further experiments exploring the photolysis of DMS oxidation products should be performed.'**

With Figure 1, to reduce the complexity of the figure, only the major reactions of the DMS pathway were included (with the more in-depth mechanism included in Appendix Table A1). In the mechanism, the photolysis of HOOCH$_2$SCH$_2$OOH does indeed break across the O-O bond in the hydroperoxy group (reaction 125), forming HOOCH$_2$SCH$_2$O. However, HOOCH$_2$SCH$_2$O then decomposes rapidly to form HOOCH$_2$S and HCHO (reaction 116). To

simplify the mechanism in Figure 1, we show this two-step process as one step, however, I am now aware that this adds confusion. To rectify this, in Figure 1 we have added an additional arrow for the photolysis of $HOOCH_2SCH_2OOH$ forming $HOOCH_2S$, to show that this is a two-step process.

Finally, the reviewer makes a good point about not including photolysis pathways consistently for all molecules. Although the mechanism does include photolysis for OCHSOH (just not included in Figure 1), we have identified that photolysis reactions have not been included for TPA and $HOOCH_2SCH_2OH$. This has now been rectified, and the simulations rerun with photolysis reactions for those two compounds in our mechanism.

Additionally, the treatment of the photolysis of the hydroperoxide group of HPMTF has been adjusted, now with the $OCH_2SCHO$ radical found decomposing to form HCHO and SCHO (which then reacts with $O_2$ to form OCS and $HO_2$), instead of reacting with $O_2$ to form OCHSCHO. This is more consistent with the experiment by Urbanski et al. (1997) which found that $CH_3SCH_2O$ quickly decomposed instead of reacting with O2.

It would help the reader to understand the mechanism better if more of an attempt were made to balance the chemical equations that are presented. There are lots of examples of this, but one such example is reaction 128.

The mechanism that is detailed in Appendix Table A1 is laid out in the same way as the MCM, where $H_2O$, $O_2$ and $CO_2$ are not included as products or reactants in the mechanism. In the case where $H_2O$ and $O_2$ are reactants, their concentration is included in the rate constant of the reaction (lines 487-493).

Reactions 132–134: Do reactions 132 and 133 ever contribute, and do reactions like this need to be included in your mechanism?

The idea behind our mechanism is to be extensive, and include reactions that may be minor or negligible in the atmosphere but could occur in chamber studies.

In this specific case, due to the reversible nature of reaction 134 ($HOOCH_2S$ + $O_2$ = HOOCH2$_2$SOO), reaction 132 ($HOOCH_2S$ + $O_3$ = $HOOCH_2SO$) is significant in both the chamber studies of Shen et al. (2022) and Jernigan et al. (2022), where HPMTF is formed ($HOOCH_2S$ is an oxidation product of HPMTF) and $O_3$ is present. Additionally, Jernigan et al. (2022) found that the reaction of $HOOCH_2S$ + $O_3$ dominated the removal of $HOOCH_2S$ in their marine box model run using their *a priori* mechanism.

---

## Author Response (AR2)

Dear Prof. Von Hobe,

Thank you for your suggestion, and for accepting the paper. We have updated the manuscript and the supplementary, changing all the y-axis labels that used ppb/ppm/ppt from 'concentration' to 'mole fraction'. Additionally, for consistency, when we refer to concentration in the text and use ppb, we've changed that to mole fraction (this was only once in the main text, and three times in the supplementary).

Thank you again for your work as editor.

Best,
Lorrie and Alex